# CoF-T2I: Video Models as Pure Visual Reasoners for Text-to-Image Generation

Chengzhuo Tong [* 1 2]   Mingkun Chang [* 3]   Shenglong Zhang [4]   Yuran Wang [1 2]   Cheng Liang [5]   Zhizheng Zhao [1]
Bohan Zeng [1 2]   Yang Shi [1 2]   Ruichuan An [1]   Yifan Dai [2]   Ziming Zhao [4]   Guanbin Li [3 6]   Pengfei Wan [2]
Yuanxing Zhang [2]   Wentao Zhang [1 †]

## Abstract

Recent video generation models have revealed the emergence of Chain-of-Frame (CoF) reasoning, enabling frame-by-frame visual inference. With this capability, video models have been successfully applied to various visual tasks (*e.g.*, maze solving, visual puzzles). However, their potential to enhance text-to-image (T2I) generation remains largely unexplored due to the absence of a clearly defined visual reasoning starting point and interpretable intermediate states in the T2I generation process. To bridge this gap, we propose **CoF-T2I**, a model that integrates CoF reasoning into T2I generation via progressive visual refinement, where intermediate frames act as explicit reasoning steps and the final frame is taken as output. To establish such explicit generation process, we curate **CoF-Evol-Instruct**, a dataset of CoF trajectories that model the generation process from semantics to aesthetics. To further improve quality and avoid motion artifacts, we enable an independent encoding operation for each frame. Experiments show that CoF-T2I significantly outperforms the base video model and achieves competitive performance, reaching 0.86 on GenEval and 7.468 on Imagine-Bench. These results indicate the substantial promise of video models for advancing high-quality text-to-image generation.

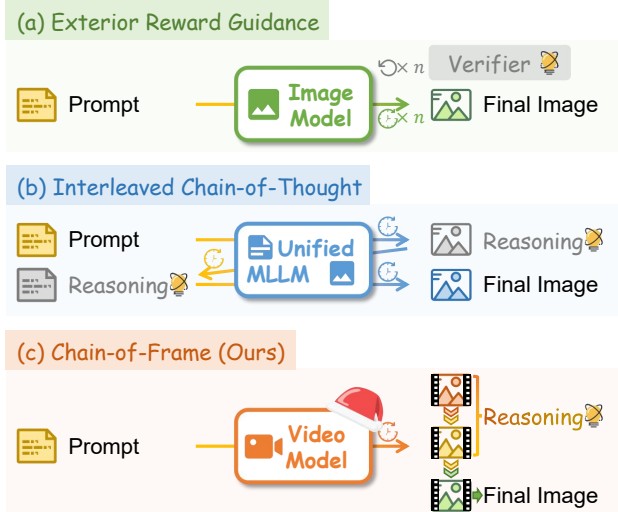

*Figure 1.* **Comparison of Inference-time Reasoning Models.** (a) Equipping image models with external verifier. (b) Interleaving textual planning within unified multimodal large language models. (c) **CoF-T2I**: Our proposed video-based CoF reasoning model.

## 1. Introduction

Recent advances in video generation models (Liu et al., 2024; Kong et al., 2025; Wan et al., 2025; Gao et al., 2025) have demonstrated the emergence of zero-shot reasoning behaviors, dubbed *Chain-of-Frame* (CoF) reasoning (Wiedemer et al., 2025). CoF reasoning leverages frame-by-frame generation to iteratively refine scenes, thereby enabling visual inference and unlocking new capabilities in perception, modeling, and manipulation. Benefiting from this, video models such as Veo3 (Google DeepMind, 2025) and Sora2 (OpenAI, 2025) have been further extended to a range of downstream visual tasks (*e.g.*, maze solving, visual puzzles) (Wiedemer et al., 2025).

Concurrently, the frontier of text-to-image (T2I) (Esser et al., 2024; OpenAI, 2024; Xie et al., 2025; Wu et al., 2025a) generation has shifted toward inference-time reasoning, primarily realized through either employing additional multimodal verifiers to assess image quality (Zhang et al., 2025a; Li et al., 2025b) or interleaving textual planning within uni-

---
[*]Equal contribution

[†]Corresponding author. [1]Peking University, Beijing, China [2]Kling Team, Kuaishou Technology, Beijing, China [3]Sun Yat-sen University, Guangzhou, China [4]Zhejiang University, Hangzhou, China [5]Nanjing University, Nanjing, China [6]Shenzhen Loop Area Institute, Shenzhen, China. Correspondence to: Wentao Zhang <wentao.zhang@pku.edu.cn>.

*Proceedings of the 43rd International Conference on Machine Learning*, Seoul, South Korea. PMLR 306, 2026. Copyright 2026 by the author(s).

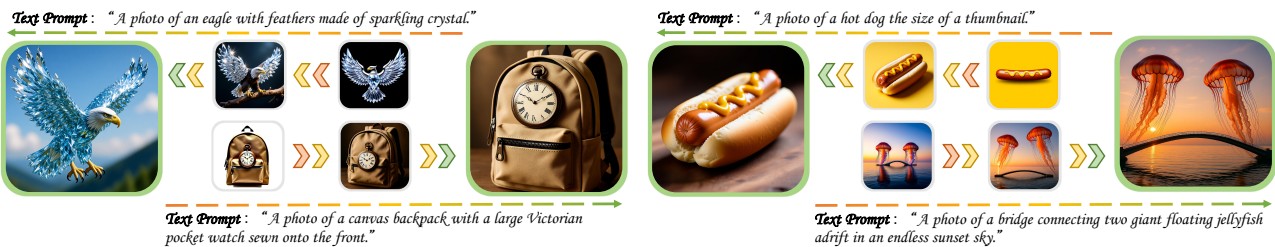

Figure 2. **Visualizations of CoF-T2I output.** We visualize the reasoning trajectories generated by CoF-T2I. For each example, the final output is shown in large, and the intermediate latent frames are shown in small.

fied multimodal large language models (MLLMs) (Jiang et al., 2025b;a; Guo et al., 2025b; Fang et al., 2025a; Xie et al., 2025; Chen et al., 2025b). However, these existing reasoning paradigms face two key limitations: first, they rely on frequent modality switching between vision and language (or scalar rewards), making pixel-level corrections indirect and lossy; second, unified MLLMs lack pretraining on large-scale, purely visual, causally ordered refinement sequences, limiting faithful frame-wise self-correction. In contrast, video models naturally model the evolution of visual states and refine scenes frame by frame under a strong spatiotemporal prior, making them particularly well-suited as pure visual reasoners to assist high-quality image generation. Still, their potential for enhancing T2I generation remains under-explored, primarily due to the lack of a clear visual reasoning starting point and interpretable intermediate states. Motivated by this, we raise a fundamental question: ***Can we use video models as pure visual reasoners to guide high-quality text-to-image generation?***

To this end, we propose ***CoF-T2I***, a model that harnesses the CoF reasoning capability of pretrained video models to enhance T2I generation. Built on a video generation backbone, CoF-T2I reimagines T2I generation as an explicit visual reasoning process. Concretely, given a text prompt, the model generates a compact three-frame sequence, where each frame represents a distinct and progressive reasoning step, from a coarse initial layout to an intermediate refinement, culminating in a high-fidelity final image. To further boost generation quality while suppressing motion artifacts inherent in video backbones, we introduce a frame-wise representation mechanism that allows the model's native video VAE to encode and decode each frame separately, thereby ensuring maximal fidelity. During inference, only the final frame undergoes full decoding and is used as the output image. We present a qualitative comparison between CoF-T2I and other inference-time reasoning models in Figure 1.

Internalizing this generation paradigm within video models necessitates vast amounts of structured visual reasoning sequences, which are largely absent from existing datasets. To fill this gap, we develop a scalable data curation pipeline and introduce ***CoF-Evol-Instruct***, a high-quality dataset com-

prising 64K CoF reasoning sequences that explicitly capture the full T2I generation process from initial semantic composition to final aesthetic refinement. Our pipeline is carefully designed to produce diverse yet consistently progressive refinement trajectories, providing clear, defect-aware supervision that guides the model from coarse, semantically noisy initial drafts toward refined, aesthetically coherent final outputs. Together, this dataset and curation pipeline enable effective training of CoF-T2I, allowing the model to internalize strong visual reasoning capabilities.

Experimental results show that CoF-T2I significantly outperforms the base video model and achieves strong performance on challenging benchmarks, attaining a competitive score of 0.86 on GenEval (Ghosh et al., 2023) and 7.468 on Imagine-Bench (Ye et al., 2025). These results highlight the substantial promise of leveraging video models' intrinsic CoF reasoning to advance high-quality T2I generation. The visualization output of CoF-T2I is shown in Figure 2.

In summary, our core contributions are as follows:

- **Video Models as Visual Reasoners:** We propose ***CoF-T2I***, a text-to-image model that unlocks the potential of video foundation models to serve as pure visual reasoners, generating images via a CoF reasoning process.

- **A comprehensive dataset with scalable pipeline:** We introduce ***CoF-Evol-Instruct***, a 64K-scale dataset of progressive visual reasoning trajectories, built with a scalable quality-aware construction pipeline.

- **Competitive results with extensive validation:** Our extensive experiments show that CoF-T2I substantially outperforms its video backbone and achieves competitive performance on challenging benchmarks, with additional validations confirming its substantial promise.

## 2. Methodology

In Section 2.1, we introduce the preliminaries of our work. In Section 2.2, we elaborate on the proposed CoF-T2I. In Section 2.3, we present our CoF-Evol-Instruct dataset as well as a detailed breakdown of the construction pipeline.

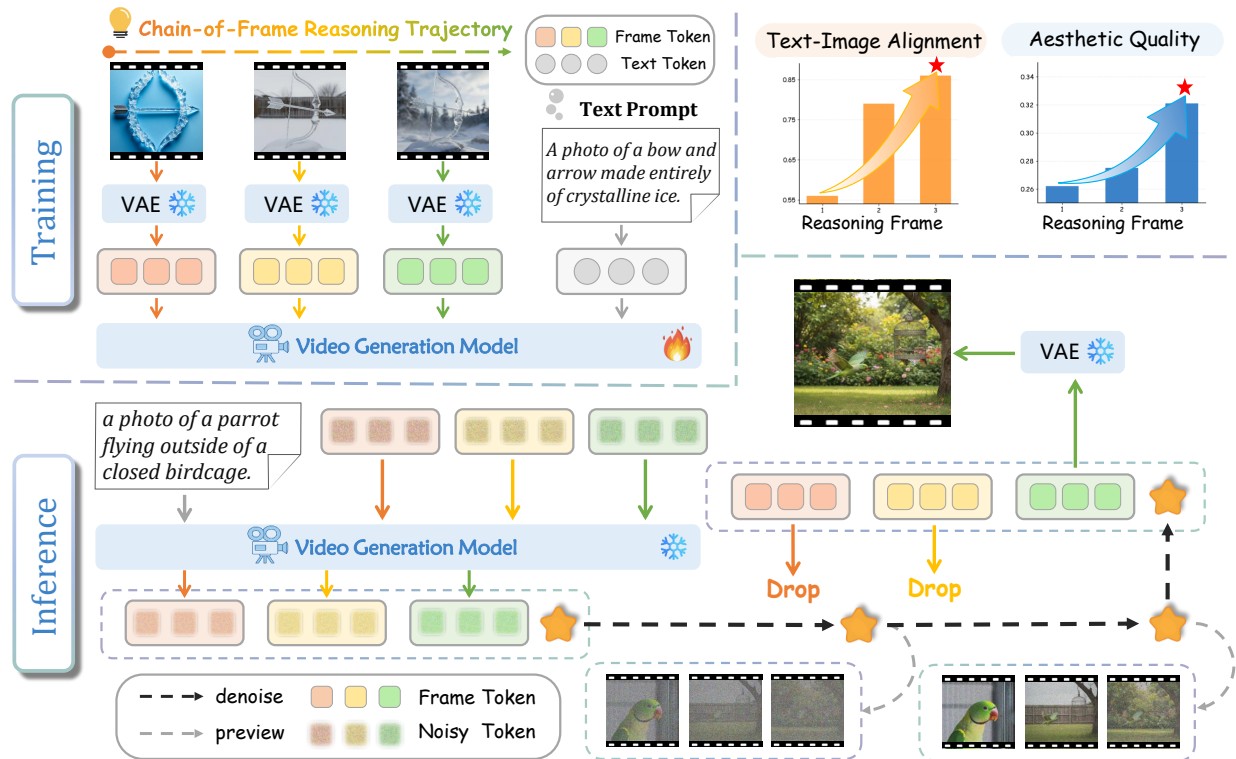

*Figure 3.* **Overview of CoF-T2I.** CoF-T2I builds on a video generation backbone, reframing inference-time reasoning for T2I generation as a CoF refinement process. **Training.** Given a CoF trajectory, we employ a video VAE to encode each frame, and optimize a vanilla flow matching objective. **Inference.** Starting from noisy initialization, the model denoises to sample a progressively refined reasoning trajectory internalized during training, only the final-frame latent is fully decoded and taken as the output image. **Quality assessment.** Along the CoF trajectory, text-image alignment and aesthetic quality continue to improve.

## 2.1. Preliminary

**Variational Autoencoder (VAE).** Modern video models typically rely on a pretrained VAE (Kong et al., 2025; Wan et al., 2025), consisting of an encoder $\mathcal{E}(\cdot)$ and a decoder $\mathcal{D}(\cdot)$, to compress video frames $V \in \mathbb{R}^{T \times 3 \times H \times W}$ into latent representation $\mathbf{z} = \mathcal{E}(V) \in \mathbb{R}^{T' \times C \times h \times w}$. We adopt the Wan2.1 VAE (Wan et al., 2025), which encodes the first frame independently, while subsequent chunks are causally compressed conditioned on prior latents. This yields $T' = 1 + \left\lfloor \frac{T-1}{4} \right\rfloor$, $h = H/8$, $w = W/8$, and $C = 16$.

**Rectified Flow.** We adopt Rectified Flow (Liu et al., 2022; Albergo & Vanden-Eijnden, 2023; Lipman et al., 2023) to model a straight path from Gaussian noise ($\epsilon \sim \mathcal{N}(\mathbf{0}, \mathbf{I})$) to video latent representations ($\mathbf{z} \sim \mathcal{E}(V)$). We train a denoiser $\mathbf{F}_\theta(\mathbf{z}_t, t; \mathbf{y})$ to predict the velocity field $\mathbf{F}_\theta(\mathbf{z}_t, t; \mathbf{y})$ given an interpolated state $\mathbf{z}_t = (1 - t)\mathbf{z} + t\epsilon$ at timestep $t \in [0, 1]$. The training objective minimizes:

$$\mathcal{L}_\theta = \mathbb{E}_{t \sim p(t), (V, y) \sim p_{\text{data}}, \epsilon \sim \mathcal{N}(\mathbf{0}, \mathbf{I})} \left[ ||\mathbf{F}_\theta(\mathbf{z}_t, t; y) - (\epsilon - \mathbf{z})||_2^2 \right] \quad (1)$$

where $\mathbf{y}$ denotes text conditions, $t \in [0, 1]$ is the timestep

sampled from a logit-normal distribution, and $p_{\text{data}}$ represents the video data distribution. This direct path learning enables high-quality and efficient generation.

## 2.2. Image Generation with Video Models

**Overview.** We propose *CoF-T2I*, a text-to-image foundation model built upon a video generation backbone, as illustrated in Figure 3. We first introduce how video models can be leveraged as CoF visual reasoners for T2I generation, then describe the frame-wise latent representation with causal VAE, and finally detail the training and inference procedure of CoF-T2I.

**Video Models as Visual Reasoners.** Video foundation models are inherently powerful visual learners and reasoners (Wiedemer et al., 2025), equipped with the natural ability to perform inference-time reasoning through Chain-of-Frame. To harness this paradigm from a robust video backbone, CoF-T2I redefines the T2I generation process as a structured refinement sequence, enabling frame-wise visual refinement to enhance T2I generation.

Formally, conditioned on a text prompt $y$, our method generates a latent sequence $\mathbf{z}_{1:3} = \{z_1, z_2, z_3\}$ that evolves from

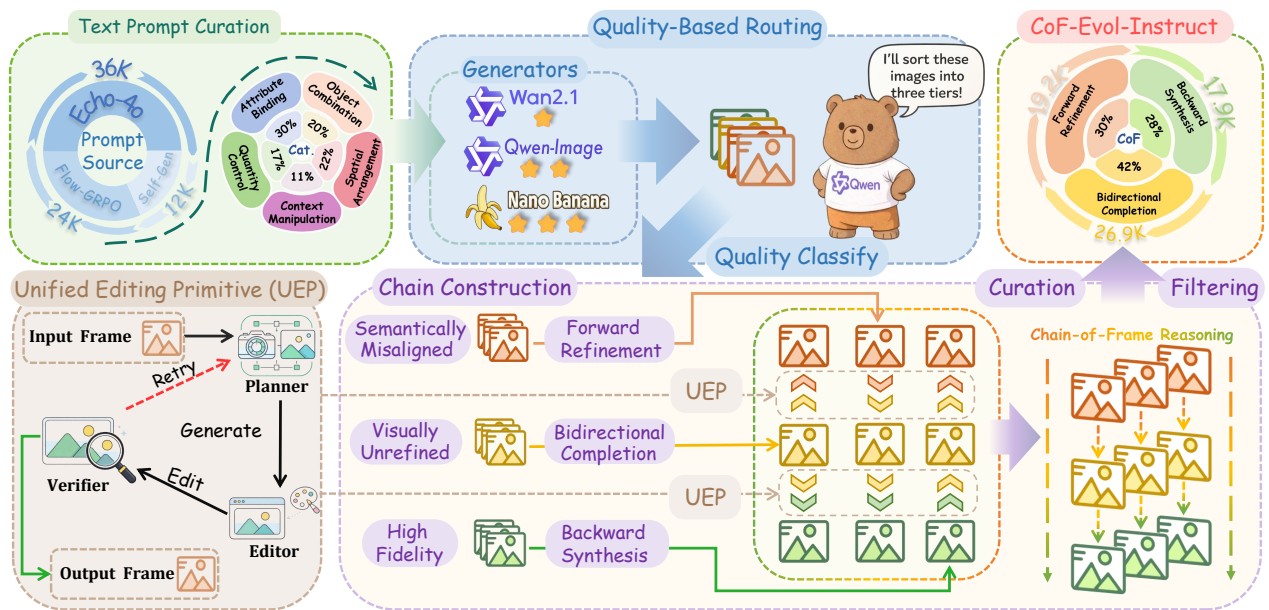

*Figure 4.* **Curation Pipeline for CoF-Evol-Instruct.** A quality-aware construction pipeline to curate reasoning data. We generate an initial pool of images across diverse distributions and dynamically route valid samples. These images are then expanded into complete CoF sequences through targeted construction strategies. Our pipeline ensures both sample-level diversity and frame-wise consistency.

coarse semantics to fine-grained aesthetics. This is achieved by modeling the joint probability distribution of the latent trajectory, conditioned by the input prompt:

$$\mathbf{z}_{1:3} \sim p_\theta(Z_{1:3}|y), \quad (2)$$

Here, $p_\theta$ is the probability density over latent sequences learned by the video model. Only the terminal latent state $z_3$, which encapsulates the culmination of the refinement, is projected into the visual space using the decoder $\mathcal{D}$ of the native causal VAE, yielding the output image:

$$\hat{I} = F_3 = \mathcal{D}(z_3). \quad (3)$$

In this way, the video model iteratively corrects artifacts and enriches details along the logical CoF trajectory, leveraging its sequence-processing architecture without extra textual planning or feedback signals.

**Frame-wise Latent Representation.** CoF-T2I adopts the causal VAE from Wan2.1 (Wan et al., 2025) to encode frames in a compact latent space. However, the native spatiotemporal compression may introduce undesired motion artifacts (*e.g.* implicit flow, dynamic inconsistencies). To mitigate this, we employ a frame-wise representation that encodes each frame independently in the latent space: We slide the VAE along the temporal axis of the video and align the input such that the target frame is always compressed within the initial context window, which spans exactly one frame. This design restricts the encoding unit to single-frame granularity, thereby preserving the spatial independence of visual features while enabling high-fidelity compression.

**Training and Inference.** During training, the model is supervised to internalize the frame-wise visual reasoning, leveraging standard flow matching objectives (Liu et al., 2022). Concretely, the model predicts the denoising targets for the latent sequence $\mathbf{z}_{1:3} = \{z_1, z_2, z_3\}$ corresponding to the CoF sequence $\{F_1, F_2, F_3\}$ and learns to produce later frames that refine earlier ones through end-to-end optimization. At inference, we generate the full latent sequence starting from Gaussian noise, effectively recovering the internalized generative reasoning via multi-step denoising. Importantly, only the final latent is fully decoded and taken as the final output image $\hat{I}$, while intermediate latents serve solely as internal states for visual reasoning.

### 2.3. *CoF-Evol-Instruct*

Constructing diverse CoF trajectories is essential for training video models to perform T2I visual reasoning, as it allows models to anticipate potential semantic defects and progressively refine aesthetic details on a semantically sound foundation. Below, we first discuss the necessity of constructing this dataset in Section 2.3.1, followed by a detailed description of our data curation pipeline in Section 2.3.2.

#### 2.3.1. NECESSITY OF DATASET CONSTRUCTION

Training CoF-T2I requires structured supervision that is both progressive and consistent, reflecting a step-wise refinement process. However, existing datasets (Ye et al., 2025; Fang et al., 2025b; Wu et al., 2025b) typically lack such visual sequences to support strict progressive refinement,

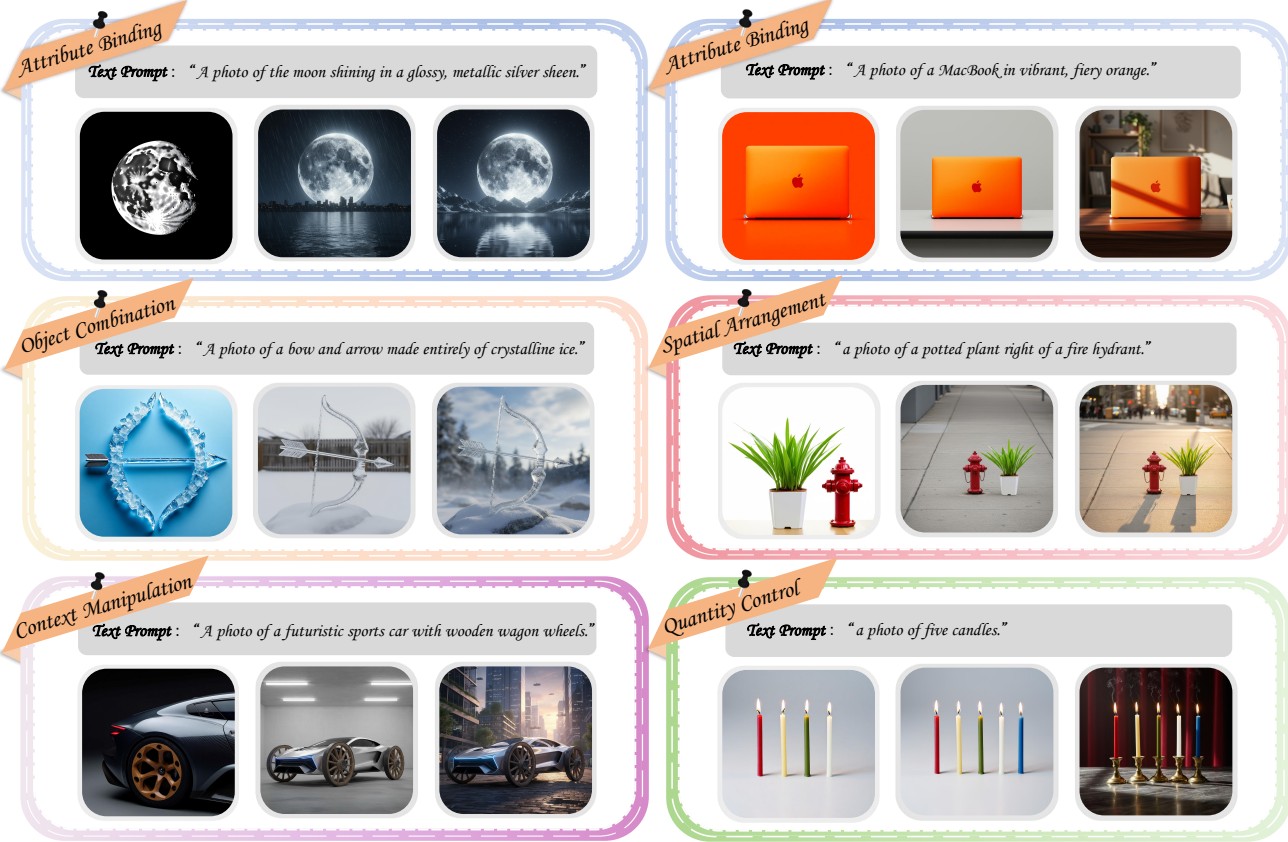

*Figure 5.* **Visualization of CoF-Evol-Instruct Dataset.** We showcase the prompt and corresponding CoF trajectories in our data, including *five* categories: *Attribute Binding*, *Object Combination*, *Spatial Arrangement*, *Context Manipulation*, and *Quantity Control*.

offering only single targets or low-quality transitions. To bridge this gap, we introduce ***CoF-Evol-Instruct***, a dataset of three-frame reasoning trajectories that capture explicit, defect-aware progression from defective initial drafts to high-fidelity final outputs. This design ensures consistent causal refinement and scalable structured supervision.

### 2.3.2. QUALITY-AWARE GENERATION PIPELINE

**Overview.** To generate CoF-Evol-Instruct at scale while maintaining both sample-level diversity and frame-wise consistency, we design a quality-aware generation pipeline. Our pipeline begins by sampling a diverse set of images from multiple T2I models. Each image is then assessed for quality and dynamically routed to one of three construction strategies, which subsequently expand it into a complete three-frame sequence: (i) forward refinement, (ii) bidirectional completion, and (iii) backward synthesis. An overview of the proposed pipeline is shown in Figure 4.

**Multi-model Sampling.** To ensure a diverse distribution of CoF sequences, we assemble a large collection of text prompts from multiple sources, covering a broad spectrum of scenes, objects, and styles. Using these prompts, we generate anchor images (*i.e.* the initial images) across several T2I models spanning different capability tiers. As a result, we obtain a heterogeneous pool of anchor images that naturally encompasses the coarse-to-fine quality spectrum. Leveraging this diversity in quality levels, we then classify each anchor image to determine the most suitable construction strategy for expanding it into a full CoF sequence.

**Quality-based Routing.** To exploit this quality diversity, we employ a vision-language model as the quality assessor, which categorizes anchors into three stages based on semantic alignment and aesthetic coherence, producing a three-way classification: *Semantically Misaligned* ($C_1$), *Visually Unrefined* ($C_2$), and *High Fidelity* ($C_3$). Here, the category index $i$ corresponds to the target position $F_i$ where the anchor is placed in the CoF sequence. Each classified anchor is then routed to the corresponding construction strategy and expanded into a complete sequence using the unified editing primitive (detailed below). This adaptive routing mechanism ensures optimal starting points for samples across the quality spectrum, thereby maximizing dataset coverage and fully utilizing anchors across all quality levels.

**Unified Editing Primitive.** To construct complete CoF

*Table 1.* **Performance comparison on GenEval.** The best and the second best *Overall* scores are in **bold** and underlined, respectively.

| Model | Single Obj. | Two Obj. | Counting | Colors | Position | Color Attr. | Overall↑ |
|---|---|---|---|---|---|---|---|
| *Standard Image Models* | | | | | | | |
| SDXL (Podell et al., 2023) | 0.98 | 0.74 | 0.39 | 0.85 | 0.15 | 0.23 | 0.55 |
| SD3-Medium (Esser et al., 2024) | 0.99 | 0.94 | 0.72 | 0.89 | 0.33 | 0.60 | 0.74 |
| FLUX.1-dev (Labs, 2024) | 0.99 | 0.88 | 0.61 | 0.87 | 0.35 | 0.55 | 0.67 |
| *Unified MLLMs* | | | | | | | |
| Janus-Pro-7B (Chen et al., 2025b) | 0.99 | 0.89 | 0.59 | 0.90 | 0.79 | 0.66 | 0.80 |
| BLIP3-o 8B (Chen et al., 2025a) | – | – | – | – | – | – | 0.84 |
| OmniGen2 (Wu et al., 2025b) | 0.99 | 0.92 | 0.77 | 0.90 | 0.82 | 0.70 | 0.80 |
| BAGEL (Deng et al., 2025) | 0.99 | 0.94 | 0.81 | 0.88 | 0.64 | 0.63 | 0.78 |
| BAGEL-Think (Deng et al., 2025) | 0.99 | 0.94 | 0.81 | 0.88 | 0.64 | 0.63 | 0.82 |
| T2I-R1 (Jiang et al., 2025a) | 0.99 | 0.91 | 0.53 | 0.91 | 0.76 | 0.65 | 0.79 |
| *Video Models* | | | | | | | |
| Wan2.1-T2V-14B (Wan et al., 2025) | 0.92 | 0.63 | 0.57 | 0.69 | 0.18 | 0.31 | 0.55 |
| **CoF-T2I (Ours)** | 0.98 | 0.95 | 0.83 | 0.89 | 0.83 | 0.71 | **0.86** |

sequences from routed anchors while ensuring cross-frame consistency and stage-specific refinement, we introduce a *unified editing primitive* (UEP) as the shared minimal operation across all strategies. UEP performs controlled, targeted edits for each stage transition (*e.g.*, semantic grounding, aesthetic refinement) while strictly preserving non-target content. We assign each prompt one of five semantic categories to narrow editing intent and improve controllability. UEP is implemented as a closed-loop system that generates minimal editing instructions, applies them, and verifies results, retrying as needed to maintain throughput. This category-conditioned primitive standardizes execution across forward, bidirectional, and backward construction, ensuring prompt-aligned and causally consistent reasoning sequences.

**Adaptive Sequence Completion.** We leverage the proposed UEP and apply three construction strategies to expand each anchor image into a complete $\{F_1, F_2, F_3\}$ sequence, ensuring progressive refinement regardless of quality stage:

- **Forward Refinement** ($F_1 \rightarrow F_2 \rightarrow F_3$). For *semantically misaligned* anchors ($C_1$), we initialize it as $F_1$ and apply forward refinement: Semantic correction via UEP to obtain a semantic-aligned $F_2$, followed by aesthetic enhancement to produce the final frame $F_3$.

- **Bidirectional Completion** ($F_1 \leftarrow F_2 \rightarrow F_3$). For *visually unrefined* anchors ($C_2$), we set the anchor as $F_2$ and expand bidirectionally: Backward via controlled semantic degradation to yield $F_1$, and forward via detailed aesthetic refinement to obtain $F_3$.

- **Backward Synthesis** ($F_1 \leftarrow F_2 \leftarrow F_3$). For *high-fidelity* anchors ($C_3$), we initialize it as $F_3$ and reconstruct backward: First apply aesthetic simplification to create $F_2$,

then introduce minimal semantic perturbation to synthesize $F_1$. All steps are executed and verified with UEP.

**Curation and Illustration.** We apply the proposed pipeline to pre-curated prompts to generate candidate reasoning chains, each paired with its corresponding prompt. After filtering out failed or incomplete samples, the curated collection, named CoF-Evol-Instruct, contains 64K high-quality CoF sequences. Representative examples spanning the five categories are presented in Figure 5. The resulting dataset provides high-quality, diverse, and progressive reasoning supervision essential for training CoF-T2I.

All implementation details, including prompt sources and counts, sampling probabilities, specific model usage, UEP process with concrete examples, and filtering criteria, are provided in Appendix B.

## 3. Experiments

### 3.1. Experimental Setting

**Evaluation.** We evaluate our proposed CoF-T2I on GenEval (Ghosh et al., 2023) and Imagine-Bench (Ye et al., 2025). GenEval is a widely-used benchmark for object-centric prompt following, focusing on composition, counting, attribute binding, and spatial relations, with automatically verifiable criteria for reliable comparison. As a complementary setting, Imagine-Bench stresses imaginative prompts that require controlled concept transformations and compositional reasoning (*e.g.*, attribute shifts and hybrid concepts), probing more abstract, concept-level semantics that go beyond literal object-centric instructions. For both benchmarks, we follow the official evaluation protocols and report the overall and category-wise scores.

*Table 2.* **Performance comparison on Imagine-Bench.** The best and the second best scores are in **bold** and underlined, respectively.

| Model | Attribute shift | Hybridization | Multi-Object | Spatiotemporal | Overall↑ |
|---|---|---|---|---|---|
| *Standard Image Models* | | | | | |
| SDXL (Podell et al., 2023) | 4.420 | 4.930 | 4.500 | 6.320 | 4.970 |
| SD3-Medium (Esser et al., 2024) | 5.140 | 6.300 | 6.070 | 5.910 | 5.780 |
| FLUX.1-dev (Labs, 2024) | 5.680 | 6.380 | 5.240 | 7.130 | 6.060 |
| *Unified MLLMs* | | | | | |
| Janus-Pro-7B (Chen et al., 2025b) | 5.300 | 6.730 | 6.040 | 7.280 | 6.220 |
| BLIP3-o 8B (Chen et al., 2025a) | 5.800 | 7.060 | 6.440 | 7.080 | 6.510 |
| OmniGen2 (Wu et al., 2025b) | 5.280 | 6.290 | 6.310 | 7.450 | 6.220 |
| BAGEL (Deng et al., 2025) | 5.370 | 6.500 | 6.410 | 6.930 | 6.200 |
| BAGEL-Think (Deng et al., 2025) | 6.260 | 7.740 | 6.960 | 7.130 | 6.930 |
| T2I-R1 (Jiang et al., 2025a) | 5.850 | 7.360 | 6.680 | 7.700 | 6.780 |
| *Video Models* | | | | | |
| Wan2.1-T2V-14B (Wan et al., 2025) | 5.436 | 6.950 | 5.383 | 6.237 | 5.939 |
| **CoF-T2I (Ours)** | 6.969 | 8.070 | 7.797 | 7.287 | **7.468** |

**Training Details.** CoF-T2I is initialized from the powerful video foundation model Wan2.1-T2V-14B and fine-tuned on our curated CoF-Evol-Instruct 64K dataset for 2 epochs, utilizing a batch size of 64, a learning rate of 1e-5, and weight decay of 1e-2. Following the strategy described in Section 2.2, we freeze the VAE and encode each frame of the sequence independently, updating only the unfrozen DiT parameters. Additionally, to mitigate the issue of incomplete subject rendering that is often encountered with default rectangular video aspect ratios (*e.g.*, $480 \times 832$), we standardize our pipeline by resizing all data to $1024 \times 1024$ during training and consistently maintaining this square resolution for inference generation.

### 3.2. Main Results

We report quantitative results in Table 1 and Table 2, respectively. In our evaluation, we benchmark CoF-T2I against two distinct categories of models: standard image models possessing solely generative capabilities, and unified multimodal models capable of leveraging textual Chain-of-Thought (CoT) for intermediate reasoning.

On GenEval, CoF-T2I achieves a leading overall score of 0.86, establishing a competitive performance among the compared methods. Notably, our pure visual reasoning approach outperforms strong unified baselines that rely on textual planning, surpassing BAGEL-Think by 0.04 and T2I-R1 by 0.07. This superior performance demonstrates the substantial advantage of our proposed CoF-T2I in generating precise and semantically correct images. Performance on Imagine-Bench further corroborates the robustness of our model. CoF-T2I delivers a remarkable improvement over the pre-trained Wan2.1 baseline, boosting the overall score from 5.939 to 7.468. Notably, the gain is especially evident

*Table 3.* **Ablation study on core mechanisms.** *Wan2.1 Base* refers to the Wan2.1-T2V-14B base model used for these comparative experiments. *Target-only SFT* fine-tunes on only the final frame of CoF-Evol-Instruct. *w/o Independent VAE* uses the default causal video-VAE encoding to encode the frame chain continuously.

| Method | Single | Two Obj. | Counting | Colors | Position | Color Attr. | Overall |
|---|---|---|---|---|---|---|---|
| Wan2.1 Base | 0.92 | 0.63 | 0.57 | 0.69 | 0.18 | 0.31 | 0.55 |
| Target-only SFT | **0.99** | 0.92 | 0.75 | 0.86 | 0.73 | 0.59 | 0.81 |
| CoF w/o Independent VAE | 0.98 | 0.94 | 0.82 | 0.84 | 0.78 | 0.63 | 0.83 |
| **CoF-T2I (Ours)** | 0.98 | **0.95** | 0.83 | 0.89 | 0.83 | 0.71 | **0.86** |

under complex composition, where the Multi-Object category reaches 7.797 compared with 5.383 for Wan2.1. This underscores the strength of CoF-T2I in handling imaginative instructions that require controlled concept transformations and complex compositional reasoning. Overall, by repurposing video foundation models as intrinsic visual reasoners, we demonstrate that this paradigm is not only viable but possesses immense potential. It offers a promising direction for future text-to-image generation, where the video model's intrinsic CoF reasoning is leveraged at inference-time to iteratively correct semantics and refine visual details, yielding higher-quality generations.

### 3.3. Ablation Study

**Does intermediate supervision yield additional benefits?** To answer this question, we examine the contribution of intermediate supervision in our Chain-of-Frame reasoning training data by comparing the full CoF-T2I model with a *Target-Only SFT* variant. For a fair comparison, CoF-T2I and *Target-Only SFT* are trained under identical settings, including the same training hyperparameters and number of optimization steps. This variant is fine-tuned using only the final frames $F_3$ from the CoF-Evol-Instruct dataset, with all intermediate reasoning frames removed during training. As illustrated in Table 3, the *Target-Only SFT* variant achieves

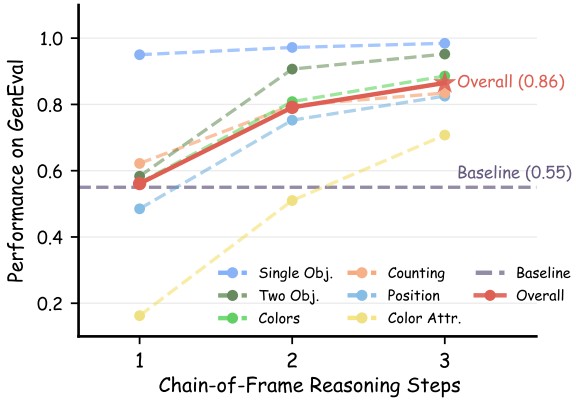 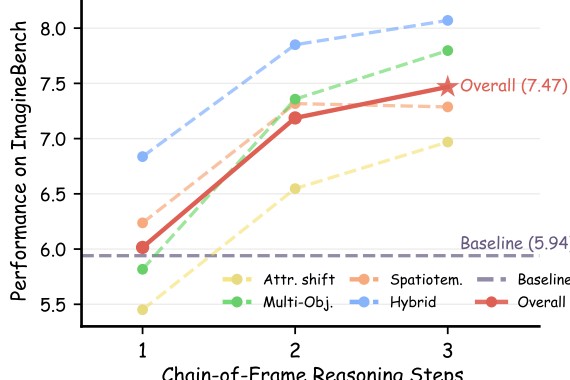

*Figure 6.* **Evolution of generation quality across reasoning steps.** We visualize the progressive improvement on both GenEval (left) and Imagine-Bench (right). The results exhibit a general ascending trend in performance scores across the inference steps.

*Table 4.* **Analysis of the Reasoning Trajectory.** We evaluate GenEval performance across each frame of the reasoning chain.

| Step | Single | Two Obj. | Counting | Colors | Position | Color Attr. | Overall |
|---|---|---|---|---|---|---|---|
| Frame 1 (Draft) | 0.95 | 0.58 | 0.62 | 0.56 | 0.49 | 0.16 | 0.56 |
| Frame 2 (Refine) | 0.97 | 0.91 | 0.80 | 0.81 | 0.75 | 0.51 | 0.79 |
| **Frame 3 (Final)** | **0.98** | **0.95** | **0.83** | **0.89** | **0.83** | **0.71** | **0.86** |

a notable improvement over the Wan2.1 base model, with the overall score rising from 0.55 to 0.81. However, it still falls short of CoF-T2I at 0.86. This disparity indicates CoF-T2I benefits not merely from stronger target supervision, but also from explicitly learning the generative trajectory, which leads to superior overall performance. We observe a consistent trend on Imagine-Bench as well, and report the corresponding ablation results in the Appendix C.

**Analysis of the Reasoning Trajectory.** To quantify how CoF inference improves generation, we evaluate each intermediate frame in the three-step reasoning chain. As shown in Table 4 and Fig. 6, GenEval performance increases monotonically from the draft $F_1$ with a score of 0.56 to the refined $F_2$ at 0.79 and further to the final $F_3$ at 0.86, with consistent gains observed across all sub-tasks. This steady progression indicates that CoF-T2I enables iterative visual self-correction, in which semantic alignment, perceptual fidelity, and visual coherence are jointly refined at each step, leading to consistent improvements across successive frames. A similar progression is also observed on Imagine-Bench, suggesting that this refinement behavior generalizes beyond object-centric evaluation settings. Detailed category-wise analyses for Imagine-Bench are provided in the Appendix C.

**Robustness Across Model Scales.** We further examine whether CoF-T2I yields consistent benefits across varying model capacities. As shown in Table 5, applying the same CoF-T2I training recipe to Wan2.1-T2V backbones at 1.3B and 14B parameters consistently improves GenEval, indicating the learned frame-evolution trajectory is effective regardless of scale. Notably, the relative gain is more pronounced

*Table 5.* **Robustness of CoF-T2I across model scales.** We report GenEval overall scores and absolute gains across 1.3B and 14B scales. *Wan2.1 Base* refers to the Wan2.1-T2V backbone.

| Model Size | Method | Overall Score | Improvement |
|---|---|---|---|
| 1.3B | Wan2.1 Base | 0.22 | - |
| | **CoF-T2I (Ours)** | **0.79** | **+0.57** |
| 14B | Wan2.1 Base | 0.55 | - |
| | **CoF-T2I (Ours)** | **0.86** | **+0.31** |

on the 1.3B model, while the 14B variant still benefits substantially. These results validate the robustness of the CoF reasoning paradigm underlying CoF-T2I, demonstrating its effectiveness across diverse model configurations.

**Necessity of Independent Frame Encoding.** We further analyze the impact of the encoding strategy in Table 3. Specifically, we compare our independent frame encoding against a *w/o Independent VAE* variant that utilizes the default causal video VAE encoding under the same training configuration. The inferior performance of this variant with a score of 0.83 suggests that treating reasoning steps as distinct and independent visual states is essential for decoupling the refined output from the defective draft. In contrast, continuous encoding might introduce unnecessary temporal dependencies that entangle reasoning steps, potentially limiting the effectiveness of visual correction. Implementation details of the continuous video VAE encoding are in the Appendix C.

## 4. Conclusion

In this work, we introduce ***CoF-T2I***, a text-to-image foundation model that repurposes pretrained video generation backbones as *pure visual reasoners*. CoF-T2I performs multi-step frame evolution at inference time, progressively correcting semantic errors and refining perceptual quality to produce higher-fidelity generations without relying on textual planning. We also curate ***CoF-Evol-Instruct*** with specialized pipelines, providing step-wise supervision signals

for progressive visual refinement. Extensive experiments on GenEval and Imagine-Bench demonstrate strong improvements over the base model and competitive performance on challenging prompt-following and compositional reasoning settings. We hope this work motivates future exploration of video-derived temporal reasoning mechanisms for more capable and controllable text-to-image generation.

## Impact Statement

This paper presents **CoF-T2I**, a text-to-image model that leverages a pretrained video model as a pure visual reasoner. It enables inference-time reasoning over structured Chain-of-Frame sequences, offering a viable alternative to conventional single-step image generation. A potential of this work is to highlight the value of exploring inference-time reasoning for text-to-image generation, using off-the-shelf video models without task-specific architectural redesign. It also offers a new perspective to investigate the reasoning capabilities of large pretrained generative models for high-quality text-to-image generation. The societal and ethical implications of our work align with those of existing text-to-image methods. Our approach introduces no new capabilities or application scenarios beyond prior generative image models, and thus raises no additional ethical concerns beyond those already addressed in existing literature.

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

# Appendix Overview

# A. Related Works

**Reasoning in Visual Generation.** While text-to-image (T2I) models like Stable Diffusion 3 (Esser et al., 2024) and FLUX.1 (Labs et al., 2025) demonstrate strong generation capabilities, their single-step generation often struggles with complex logical reasoning. Recent studies (Snell et al., 2024; Ma et al., 2025) indicate that scaling test-time computation effectively improves performance. Building on this, numerous works propose employing additional models to guide image generation (Zhang et al., 2025a; Li et al., 2025b; Zhuo et al., 2025). For instance, Image-Gen-CoT (Zhang et al., 2025a) introduces PARM as a reward model for test-time verification, while ReflectionFlow (Zhuo et al., 2025) provides natural language feedback via external vision-language models. Unlike external-signal-reliant approaches, another paradigm (Zhou et al., 2024; Wu et al., 2024; Xie et al., 2025; Chen et al., 2025b; Deng et al., 2025) uses a unified multimodal large language model to decompose single-step generation int interleaved textual reasoning and visual synthesis (Jiang et al., 2025a; Huang et al., 2025; Chern et al., 2025; Guo et al., 2025b; Jiang et al., 2025b). Specifically, TwiG (Guo et al., 2025b) interleaves textual reasoning for on-the-fly guidance and Draco (Jiang et al., 2025b) explores a path of pure visual reasoning, utilizing pre-planned visual drafts to guide the final generation.

**Zero-Shot Reasoning in Video Models.** With the rapid development of video generation models (Liu et al., 2024; Kong et al., 2025; Wan et al., 2025; Gao et al., 2025), new possibilities for pure visual reasoning have emerged. The step-by-step reasoning paradigm, originally pioneered by Chain-of-Thought (CoT) in language models (Wei et al., 2023; Kojima et al., 2023; Shao et al., 2024), has recently inspired visual analogs (Li et al., 2025a; Rotstein et al., 2025). Chain-of-Frame (CoF) reasoning, first introduced in (Wiedemer et al., 2025), demonstrated that video models can tackle complex visual tasks by recasting problems into spatiotemporal progressions, where each frame serves as an intermediate step of visual inference, analogous to CoT's intermediate reasoning tokens. Building on this, a growing body of work (Guo et al., 2025a; Yang et al., 2025; Luo et al., 2025; Zhang et al., 2025b; Team et al., 2025; Deng, 2025) has systematically evaluated the emergent CoF capability, quantifying video models' strengths and limitations across spatial, physical, and logical reasoning tasks. These studies collectively suggest that CoF holds strong potential for complex visual reasoning, frequently demonstrating advantages over approaches that rely on language mediation.

# B. Dataset Construction Details

We curate 68K unique text prompts by assembling 24K from (Liu et al., 2025), 37K from (Ye et al., 2025), and 12K self-generated prompts adapted from (Ghosh et al., 2023), followed by prompt-level deduplication. Anchor images are generated using T2I models across three capability tiers: Wan2.1 (Wan et al., 2025) (weak), Qwen-Image (Wu et al., 2025a) (medium), and Nano-Banana (Sharon & Brichtova, 2025) (strong). Tiers are sampled with probabilities $p_{\text{weak}} = 0.25$, $p_{\text{medium}} = 0.5$, $p_{\text{strong}} = 0.25$, yielding 68K anchors that span a heterogeneous coarse-to-fine quality spectrum.

## B.1. Details on Prompt Categorization

To make semantic-stage edits in UEP targeted and controllable, we categorize each prompt by its *primary constraint* (*i.e.*, the dominant factor that must be changed to satisfy the prompt while keeping other content invariant). The five categories are:

- **Attribute Binding.** Constraints binding intrinsic attributes (*e.g.*, color, shape/size, material) to an object. Object identity and scene remain unchanged; only the specified property is modified.

- **Object Combination.** Constraints involving multi-object composition or creative hybrids (*e.g.*, unusual juxtapositions, hybrid concepts). Focus is on object-level interactions.

- **Quantity Control.** Constraints requiring specific instance counts or count-sensitive descriptions. The primary variable is object quantity.

- **Spatial Arrangement.** Constraints specifying relative spatial relations among objects (*e.g.*, left/right, above/below, in/on, front/behind). Object identities are preserved; satisfaction depends on correct configuration.

- **Context Manipulation.** Constraints dominated by global scene-level context (*e.g.*, temporal cues, environmental settings, background/location) that cannot be reduced to single-object edits.

## B.2. Implementation Details

**Quality-based Routing.** We employ Qwen3-VL-8B (Bai et al., 2025) to perform quality-based routing for anchor images. Given a prompt and generated image, the model classifies it into one of three stages: *Semantically Misaligned* $(C_1)$ (major semantic violations regardless of aesthetics),

*Visually Unrefined ($C_2$)* (semantically correct but aesthetically low-quality, *e.g.*, lacking texture/lighting/realism), or *High Fidelity ($C_3$)* (satisfying both semantic alignment and aesthetic coherence). The classification prompt template is:

---

**Quality Assessing Prompt Template**

```
You are a strict image quality assessor.
Input:
- PROMPT: {prompt}
- IMAGE: [Image Input]
Evaluate based on Semantic Alignment and
Aesthetic Quality:
1.  Semantically Misaligned (C1):  Major
semantic identity errors (e.g., wrong
objects), regardless of aesthetics.
2.  Visually Unrefined (C2):
Semantically correct, but low aesthetic
quality (blur, distortion, bad
lighting).
3.  High Fidelity (C3):  Semantically
correct AND high aesthetic quality.
Output Format (JSON-compatible):

{
   "label": "C1" | "C2" | "C3",
   "analysis": "strict reasoning based on
   the definitions above",
}
```

---

**Unified Editing Primitive (UEP).** UEP is the core modular operation enabling consistent, category-conditioned stage transitions across all construction routes. It is implemented as a closed-loop agent system with three components powered by Qwen-family models: planner and verifier (Qwen3-VL-32B (Bai et al., 2025)), and editor (Qwen-Image-Edit-2509 (Wu et al., 2025a)). The planner generates minimal editing instructions, the editor applies them while preserving non-target content, and the verifier outputs binary success $b \in \{0, 1\}$ (retrying up to $K = 3$ times). UEP is conditioned on one of the five semantic categories listed above to narrow editing intent and improve controllability.

**Prompt Designs.** To achieve precise control, we design role-specific system prompts for every component of UEP for planner and verifier. We offer simplified templates:

---

**Planner Prompt Template**

```
You are a visual reasoning planner for
image editing.
Input Context:
- CURRENT IMAGE: [current frame]
- PROMPT: {prompt}
- STAGE: {stage} (semantic correction OR
aesthetic refinement)
- DIRECTION: {direction} (forward OR
backward)
- CATEGORY: {category} (one of
[Attribute Binding, etc.])
```

---

```
- Edit Hint:  {hint} (based on CATEGORY)
- PREVIOUS FRAME: [prior frame,
optional]
Editing Logic:
Generate a targeted editing instruction.
If DIRECTION is forward, fix semantic
discrepancies in the CATEGORY or enhance
aesthetic realism.  If DIRECTION is
backward, explicitly introduce semantic
errors related to the CATEGORY or
degrade the visual quality.
Task:
Output exactly ONE minimal editing
instruction (< 40 words) that achieves
the goal derived above.  If the PREVIOUS
FRAME is provided, use it to maintain
subject identity strictly.
Output Format (JSON-compatible):

{
   "instruction": "edit instructions",
}
```

---

**Verifier Prompt Template**

```
You are a visual reasoning verifier for
image editing chains.
Input Context:
- EDITED IMAGE: [new frame]
- PROMPT: {prompt}
- STAGE: {stage} (semantic correction OR
aesthetic refinement)
- DIRECTION: {direction} (forward OR
backward)
- CATEGORY: {category} (one of Attribute
Binding, Object Combination, Quantity
Control, Spatial Arrangement, Context
Manipulation)
- PREVIOUS FRAME: [prior frame,
optional]
Verification Logic:
Verify transition matches DIRECTION:
forward improves semantics/aesthetics;
backward degrades or introduces errors.
Ensure main subject identity consistency
with PREVIOUS FRAME and no unrelated
artifacts.
Task:
Output SUCCESS(1) if goal achieved
while preserving unrelated content;
FAIL(0) otherwise (imperceptible change,
over-editing, wrong direction).
Output Format (JSON-compatible):

{
   "Output": 0 OR 1,
}
```

---

**Workflow and Resolution Strategy.** UEP follows a 'planner → editor → verifier' iterative loop. The planner produces a concise instruction, the editor executes it, and the verifier checks the outcome. If verification fails after $K = 3$ retries, the pipeline falls back to direct regeneration using

*Table 6.* **Performance on Imagine-Bench.** We report per-type average scores and the overall weighted score. *Wan2.1 Base* denotes the Wan2.1-T2V-14B base model without CoF fine-tuning. *Target-only SFT* is fine-tuned using only the final frames of CoF-Evol-Instruct.

| Method | Attribute Shift | Hybridization | Multi-Object | Spatiotemporal | Overall |
|---|---|---|---|---|---|
| Wan2.1 Base | 5.436 | 6.950 | 5.383 | 6.237 | 5.939 |
| Target-only SFT | 5.940 | 7.540 | 7.220 | 6.727 | 6.755 |
| **CoF-T2I (Ours)** | **6.969** | **8.070** | **7.797** | **7.287** | **7.468** |

Qwen-Image (Wu et al., 2025a) to maintain data quality and throughput. To balance efficiency and precision, we use resolution-adaptive perception: $512 \times 512$ for F1 $\leftrightarrow$ F2 transitions (sufficient for semantic correctness), and full $1024 \times 1024$ for F2 $\leftrightarrow$ F3 (better assessment of fine-grained aesthetic details).

**Construction Strategy Examples.** The three strategies ensure progressive refinement regardless of starting quality:

- **Forward Refinement** ($F_1 \rightarrow F_2 \rightarrow F_3$): For semantically misaligned anchors ($C_1$), We initialize it as $F_1$ and apply UEP for semantic correction (*e.g.*, fix missing objects, incorrect attribute bindings, implausible layouts) to $F_2$, then aesthetic enhancement to $F_3$.

- **Bidirectional Completion** ($F_1 \leftarrow F_2 \rightarrow F_3$): For visually unrefined anchors ($C_2$), we set it as $F_2$ and perform backward controlled semantic degradation (*e.g.*, weakening attributes, dropping secondary objects) to yield $F_1$; forward aesthetic refinement then focuses on high-frequency details and visual harmony to reach $F_3$.

- **Backward Synthesis** ($F_1 \leftarrow F_2 \leftarrow F_3$): For high-fidelity anchors ($C_3$), we initialize it as $F_3$ and conduct backward aesthetic simplification (*e.g.*, reduced sharpness, softened lighting complexity) to create $F_2$, followed by minimal category-conditioned semantic perturbation (*e.g.*, altering count, degrading specific attributes) to synthesize a coarse $F_1$.

All steps are executed and verified with UEP for prompt consistency and inter-frame coherence.

## C. More Experiment Details

In this section, we provide additional results on Imagine-Bench and implementation details regarding the continuous video VAE encoding and the system prefix used during training and inference.

### C.1. Additional Results on Imagine-Bench

**Target-only SFT.** We evaluate the *Target-only SFT* variant on Imagine-Bench to assess the performance when the model is fine-tuned solely on the final output frames, without intermediate reasoning supervision. The per-type average scores are reported in Table 6.

*Table 7.* **Analysis of the Reasoning Trajectory on Imagine-Bench.** We report per-type scores for each frame of the visual reasoning chain, illustrating progressive refinement from the draft to the final output.

| Step | Attribute Shift | Hybridization | Multi-Object | Spatiotemporal | Overall |
|---|---|---|---|---|---|
| Frame 1 (Draft) | 5.451 | 6.837 | 5.817 | 6.237 | 6.015 |
| Frame 2 (Refine) | 6.547 | 7.850 | 7.357 | **7.317** | 7.187 |
| **Frame 3 (Final)** | **6.969** | **8.070** | **7.797** | 7.287 | **7.468** |

**Analysis of the Reasoning Trajectory.** We further analyze the reasoning trajectory of CoF-T2I on Imagine-Bench by evaluating the generation quality at each step of the chain ($F_1 \rightarrow F_2 \rightarrow F_3$). As shown in Table 7, scores consistently improve across all categories as the reasoning progresses from the initial draft to the final output.

### C.2. Implementation Details

**Continuous Video VAE Encoding.** Standard causal video VAEs typically adopt a spatiotemporal compression schedule of $1 + 4n$: the first frame is encoded without temporal downsampling, while subsequent frames are temporally compressed by a factor of $4$. As a result, a 3-frame chain ($F_1, F_2, F_3$) is not directly compatible with the native temporal layout. To enable continuous video VAE encoding, we pad each training sample to five frames by repeating the final frame:

$$(F_1, F_2, F_3) \rightarrow (F_1, F_2, F_3, F_3, F_3).$$

The padded 5-frame clip is then passed through the native video VAE encoder. During decoding, we only decode and retain the *last frame* as the final output image, while all other decoded frames are discarded. For text-to-image generation, only the last decoded frame is used as the model output.

**System Prompt Prefix.** To enable the Chain-of-Frame reasoning capability, we append a fixed system prefix to the prompt during both training and inference. This prefix instructs the model to generate a short refinement chain that preserves concept and composition while improving quality step by step. The specific prefix used is:

> **System Prompt Prefix**
>
> ```
> Generate a short refinement chain of the
> same concept and composition, improving
> the image step by step.
> Prompt: <user prompt>
> ```

## D. Limitations and Future Work

While CoF-T2I demonstrates the efficacy of leveraging video models' Chain-of-Frame reasoning for text-to-image generation, our study has not systematically explored its extension to broader task domains, such as video-related tasks (*e.g.*, text-to-video generation) or 3D-related tasks (*e.g.*, text-to-3D synthesis). Extending to text-to-video, for

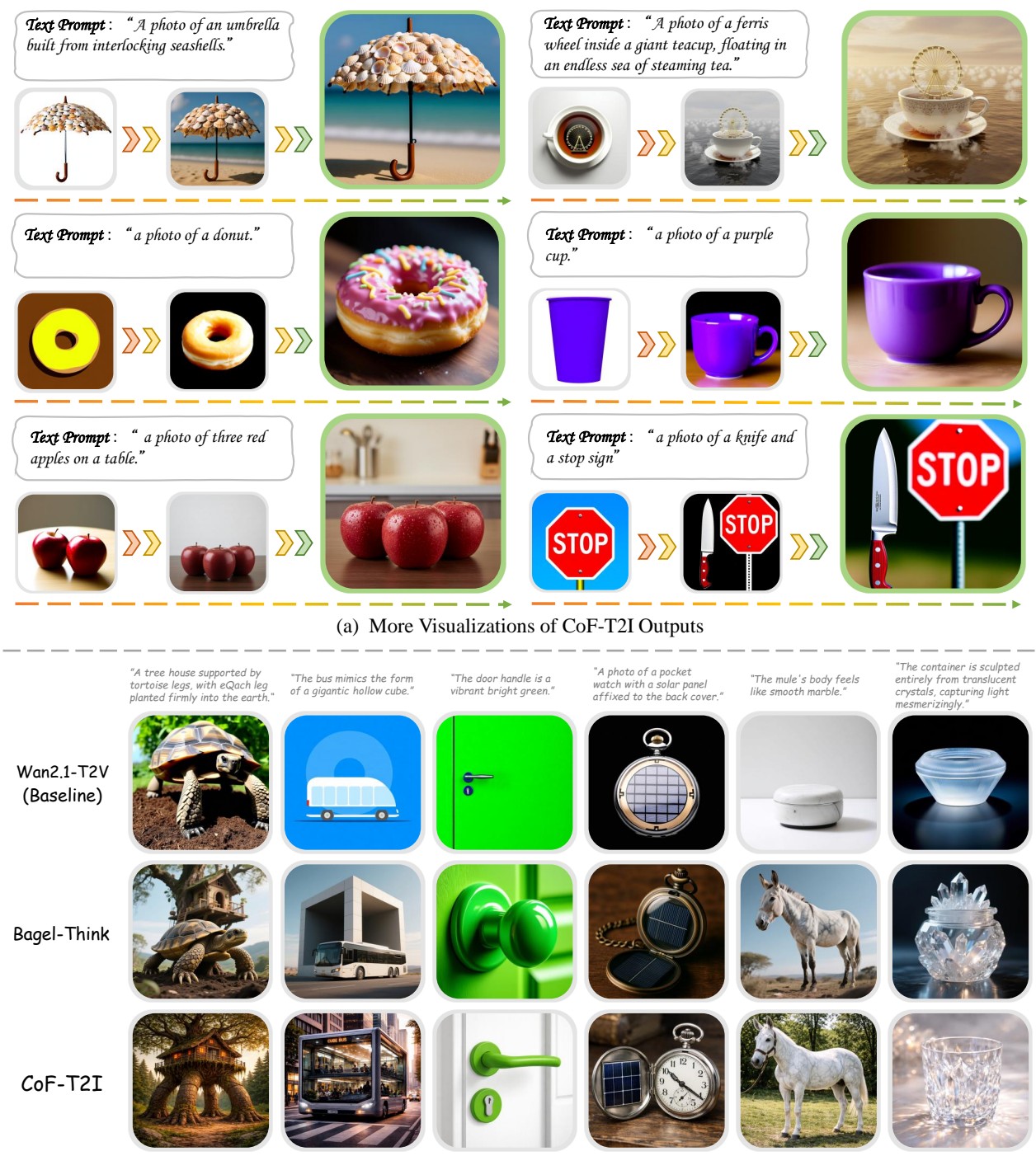

(a) More Visualizations of CoF-T2I Outputs

(b) Comparison with Other Models

*Figure 7.* **Qualitative Analysis: Reasoning Trajectories and Comparisons for *CoF-T2I***

instance, could introduce challenges like handling longer temporal sequences, increased computational demands for multi-frame refinement, maintaining dynamic coherence without introducing unintended motion artifacts, and sourcing high-quality datasets for extended reasoning chains. Ad-

ditionally, reinforcement learning (RL) techniques, which have proven successful in enhancing Chain-of-Thought reasoning in textual domains through iterative feedback and reward optimization, remain underexplored in our framework; integrating RL with our video-based model could

enable more adaptive visual refinements in T2I tasks, potentially improving robustness to diverse prompts and further elevating output quality. Future work will investigate these avenues to unlock the full potential of video foundation models as versatile visual reasoners.

## E. Qualitative Examples

We provide further qualitative visualizations and comprehensive comparisons in Figure 7. In Figure 7(a), we showcase the complete reasoning trajectory, including the intermediate frames and the final output alongside their corresponding prompts. The examples encompass diverse generation scenarios, such as imaginative object combination, specific attribute binding, and spatial arrangement. By visualizing the chain-of-frame evolution, we demonstrate how the model iteratively refines semantics or reconstructs details to achieve the final target.

In Figure 7(b), we compare our method with the baseline video model, Wan2.1-T2V-14B, and a representative inference-time reasoning model, BAGEL-Think, which interleaves textual Chain-of-Thought with visual generation. As shown, the baseline Wan2.1-T2V tends to rely heavily on training priors and often ignores counter-intuitive instructions; for instance, it generates a standard bus shape instead of the requested "gigantic hollow cube" and fails to render the "vibrant bright green" door handle. While Bagel-Think exhibits better prompt following than the baseline, it still struggles with fine-grained texture synthesis and complex structural deformations, such as the "marble" texture on the mule or the "translucent crystal" container. In contrast, CoF-T2I produces satisfying results with both high photorealistic quality and precise alignment with the prompt, successfully executing challenging instructions that require strong reasoning capabilities.

Collectively, these results suggest that CoF-T2I improves upon prior paradigms by leveraging spatiotemporal priors of video models, eliminating the need for textual intermediaries and external reward guidance. By employing visual CoF reasoning, our design shows potential for robustly generating complex structures, offering a compelling alternative to approaches reliant on modality switching.

