# OpenReview forum: "CoF-T2I: Video Models as Pure Visual Reasoners for Text-to-Image Generation"
_ICML.cc/2026/Conference — ICML 2026 regular_

### Official Review · Reviewer_2zMK · 2026-02-20

**Soundness:** 2
**Presentation:** 2
**Significance:** 2
**Originality:** 1
**Overall Recommendation:** 3
**Confidence:** 4

**Summary:**

This paper presents **CoF-T2I**, which repurposes a pretrained **text-to-video** foundation model for **text-to-image generation** by generating a short **three-frame “Chain-of-Frame” refinement trajectory** (coarse → detailed) and returning only the final frame. To reduce video-VAE motion artifacts, it decodes frames independently, and trains the model with a flow-matching objective. The authors also introduce **CoF-Evol-Instruct**, a **64K** dataset of curated three-frame refinement trajectories for supervision. Experiments report strong text-to-image performance (e.g., GenEval 0.86, Imagine-Bench 7.468) and show improvements across intermediate frames.

**Compliance With Llm Reviewing Policy:**

Affirmed.

**Final Justification:**

The paper is clearly written, but its novelty is limited. The pipeline relies heavily on an existing T2I model and incurs higher costs due to its dependence on priors. Overall, its theoretical and methodological contributions fall short of ICML standards, though the empirical results are strong, placing it at a borderline level.

**Key Questions For Authors:**

1. **Ablation of contributions:** How much of the gain comes from (i) the 3-frame CoF formulation, (ii) independent per-frame VAE decoding, and (iii) CoF-Evol-Instruct supervision, respectively? Clear ablations isolating each factor would materially affect my assessment of soundness and originality.
2. **Sensitivity to chain length:** Why is a 3-frame chain sufficient, and how do results change with 1/2/4/6 frames under a comparable compute budget? If performance is robust and plateaus early, it strengthens the method; if heavily tuned, it weakens generality.
3. **Backbone dependence and transfer:** Does CoF-T2I generalize across different T2V backbones (not only Wan2.1), and across domains/resolutions? Evidence of transfer would significantly strengthen the significance claim.
4. **Dataset bias and transparency:** What are the filtering/routing criteria in CoF-Evol-Instruct, and how do you ensure diversity and avoid benchmark leakage? If the dataset construction is robust and well-controlled, it strengthens confidence that improvements are not artifact-driven.

**Limitations:**

The paper would benefit from a clearer limitations discussion in at least the following aspects:

- Backbone and compute dependence. Since the approach relies on a strong pretrained T2V foundation model and a multi-frame refinement trajectory, the paper should explicitly discuss runtime/memory overhead relative to standard T2I diffusion and how performance degrades with smaller/weaker backbones.

- Dataset construction bias. CoF-Evol-Instruct is curated via routing/editing, which can introduce stylistic bias or benchmark-specific artifacts; the paper should more explicitly describe potential biases, diversity trade-offs, and failure cases.

**Strengths And Weaknesses:**

## Strengths

- The proposed “Chain-of-Frame” formulation is technically coherent: using a short multi-frame refinement trajectory to progressively improve a single final image is a reasonable way to reuse a T2V backbone for T2I, and the independent per-frame VAE decode is a pragmatic design choice to mitigate video-specific artifacts.
- The paper’s main idea is easy to understand and the overall pipeline (three-frame coarse-to-fine refinement, output last frame) provides a clear narrative that connects motivation, method, and evaluation.

## Weaknesses

- It is not fully clear whether gains come primarily from (i) the CoF trajectory itself, (ii) the new supervision data, or (iii) architectural details like per-frame VAE decoding; stronger ablations would be important to isolate each factor.
- CoF-Evol-Instruct is curated through routing/editing; such pipelines can introduce stylistic or benchmark-specific biases that may inflate measured improvements or reduce diversity. More transparency on filtering criteria and failure cases would help.
- While the framing is appealing, the method may be viewed as an inference/training re-parameterization of existing diffusion refinement ideas; the paper would benefit from deeper analysis explaining *why* three-frame CoF is sufficient and when/why it fails.

---

> ### Author Rebuttal · Authors · 2026-03-31
>
> We sincerely appreciate your comments. We provide comprehensive statistics on our [*[anonymous webpage](https://anonymous.4open.science/w/vm433F8/)*], and respond to each comment in detail below:
>
> ---
> > #### **Q1: Contribution Attribution and Source of Gains.**
>
> >
> To isolate the source of gains, we conducted a rigorous ablation study, with quantitative results provided on the webpage (**Table A**):
> * **Target-only SFT (Sec. 3.3):** Fine-tuning only on final frames achieves 0.81 on GenEval, 0.05 lower than full CoF-T2I, showing the benefit of explicit multi-step reasoning over single-step generation.
> * **Independent VAE decoding (Appendix C.2)**: Reverting to continuous temporal encoding lowers GenEval by 0.03, validating the need to isolate visual states.
> * **Dataset Supervision:** Removing the ***progressive structure*** (repeating final frames as intermediates; 0.80 on GenEval) or the ***causal ordering*** (randomly permuting intermediates; 0.77 on GenEval) leads to clear degradation. This shows that reasoning capability relies on structured trajectory supervision.
>
> Importantly, ***replaying the curation pipeline at inference time*** yields only 0.83 on GenEval (see ***Reviewer Vuw3 (Q1)***). This gap demonstrates CoF-T2I learns a generalizable reasoning prior that exceeds raw data curation.
>
> ---
> > #### **Q2: Sensitivity to Chain Length and Limitations.**
>
> >
> **Chain Length Sensitivity**: We conduct analyses and ablations on trajectory length (see ***Reviewer 34vG (Q1)***). Training with 1/2/3 frames yields progressive GenEval gains of 0.81/0.83/0.86. Extending inference to 4/6 frames causes performance to plateau at 0.87/0.86 while latency increases linearly (from 121.70s to 300.87s), validating 3 frames as an effective reasoning decomposition.
>
> **Limitations**: CoF may encounter the following issues:
> * ***Complex spatial overlaps***: Prioritizing local coherence over global layout.
> * ***Early detail convergence***: High-fidelity early drafts resist semantic shifts.
>
> ---
> > #### **Q3: Dataset Bias, Transparency, and Leakage.**
>
> >
> We provide full pipeline statistics and filtering details on the webpage (**Table B**).
>
> 1. **Biases & Diversity Constraints:** Validation with Qwen3-VL-32B exhibits stylistic bias (e.g., specific lighting). The UEP process inherently limits global layout diversity in a single trajectory. We counter this by diversifying prompt sources and sampling models (Appendix B).
>
> 2. **Failure Cases:** Typical failures include ***(1)*** subject identity shifts during detailed editing and ***(2)*** unintended background changes.
>
> 3. **Leakage Mitigation:** We further evaluate ***embedding similarity*** between training and test sets using all-MiniLM-L6-v2 [1]. Only 4.1% and <0.1% of samples exceed 0.85 and 0.95 thresholds, indicating negligible semantic leakage.
>
> [1] Making Monolingual Sentence Embeddings Multilingual using Knowledge Distillation. ACL 2020.
>
> ---
> > #### **Q4: Generalization, Compute and Capacity.**
>
> >
> **Robust Generalization:** We provide quantitative results on the webpage (**Table C, D, E**).
> * ***Backbones***: Applying CoF-Evol-Instruct recipe to **HunyuanVideo-1.5** [1] improves GenEval by 0.33, confirming model-agnostic reasoning priors.
> * ***Domains***: CoF-T2I maintains robust out-of-domain generalization on **TIIF-Bench** [2], yielding competitive performance against strong baselines (e.g., BAGEL).
> * ***Resolutions***: We evaluate CoF-T2I across multiple inference resolutions (512px to 1280px). Sustained improvements along the chain show robust spatial generalization.
>
> **Compute Overhead and Capacity:** We evaluate performance and compute across backbones/scales (see ***Reviewer 34vG (Q2)***). CoF-T2I-1.3B outperforms FLUX.1-dev with comparable latency, while CoF-T2I-14B yields a 0.713 Imagine-Bench gain over single-step model (Target-only SFT), with additional 18.3% latency overhead vs. textual CoT (BAGEL-Think). This confirms that visual reasoning optimizes test-time compute for better compositional fidelity.
>
> [1] HunyuanVideo 1.5 Technical Report. arXiv 2025.
> [2] TIIF-Bench: How Does Your T2I Model Follow Your Instructions? arXiv 2025.
>
> ---
> > #### **Q5: Differences from Diffusion Refinement.**
>
> >
> We clarify that CoF-T2I is not a re-parameterization of diffusion refinement:
>
> 1. **Semantic Reasoning vs. Pixel Denoising:** Standard refinement improves pixel fidelity within a single continuous denoising path. In contrast, CoF-T2I leverages ***video spatiotemporal priors*** to model discrete logical steps across frames, semantically correcting flawed layouts.
>
> 2. **Trajectory Supervision Efficacy:** If CoF-T2I were merely an inference reparameterization, intermediate structure would be irrelevant. However, CoF-T2I significantly outperforms the Target-only SFT baseline on GenEval (0.86 vs 0.81) and Imagine-Bench (7.468 vs 6.755). This proves the model internalizes a genuine ***generative reasoning process***, rather than just memorizing a stronger target distribution.

---

> > ### Author Rebuttal · Reviewer_2zMK · 2026-04-03
> >
> > The author provides several experiments and analyses to answer my question. But due to the complexity of the pipeline and the innovation of contributions, I decided to keep my score.

---

> > > ### Author Response · Authors · 2026-04-04
> > >
> > > Thank you for reviewing our rebuttal and keeping the discussion open! We apologize for any ambiguity that may have obscured the simplicity and core novelty of our framework. To clarify: Our pipeline is not a complex heuristic, but an ***automated, data-efficient, and scalable*** system. Furthermore, our core innovation lies in repurposing ***video models*** to introduce a ***new test-time compute paradigm*** for explicit, stepwise visual reasoning in T2I generation.
> > >
> > > ---
> > > > #### **Q1: Complexity of the Pipeline.**
> > >
> > > >
> > > While the manuscript describes our data curation in detail, the underlying logic of our pipeline is quite straightforward:
> > >
> > > 1. **Automated, Scalable, and Data-Efficient Curation.** Our pipeline operates on a highly intuitive closed-loop framework designed for two key properties: ***(1) Universal Inclusivity.*** Instead of discarding suboptimal generations, our pipeline dynamically accommodates inputs across a broad quality spectrum, ranging from coarse layouts to high-fidelity images. By adaptively assigning the appropriate expansion strategy, every sample is effectively utilized, demonstrating strong sample efficiency. ***(2) Effortless Scalability.*** Benefiting from the routing mechanism, our pipeline automates the trajectory expansion without requiring any human-in-the-loop annotation or manual intervention, rendering it inherently scalable. This enables us to seamlessly produce diverse, high-quality reasoning trajectories directly from raw model outputs.
> > >
> > > 2. **Empirical Necessity of the Pipeline.** Our pipeline is critical for achieving our results. If generating step-by-step reasoning trajectories were redundant, simply fine-tuning the model on the high-fidelity final frames (F3) would yield comparable performance. However, our ablation study shows that the ***Target-only SFT*** variant (detailed in ***Sec. 3.3 of the paper***) achieves only 0.81 on GenEval and 6.755 on Imagine-Bench. In contrast, training on the complete reasoning trajectories synthesized by our pipeline (full CoF-T2I) significantly boosts performance to 0.86 and 7.468, respectively. This large gap verifies that the structured intermediate steps from our pipeline enable the model to effectively learn the process of visual reasoning, rather than merely memorizing high-quality outputs. Furthermore, these curated trajectories provide a robust, model-agnostic prior. As shown in our previous response, applying our data recipe to ***HunyuanVideo-1.5*** yields a substantial GenEval gain from 0.51 to 0.84, showing the value and generalizability of our pipeline's output.
> > >
> > > ---
> > > > #### **Q2: Innovation of Contributions.**
> > >
> > > >
> > > Our fundamental innovation is to propose a paradigm shift: repurposing ***video models*** as pure visual reasoners for text-to-image generation.
> > >
> > > 1. **Eliciting Video-based Reasoning Priors.** Our core contribution lies in unveiling an efficient path to advanced T2I visual reasoning: rather than building T2I models from scratch, we demonstrate the viability of ***repurposing existing video foundation models***. These models are trained on massive causal visual sequences, enabling them to encode strong zero-shot spatiotemporal reasoning priors and rich world knowledge [1]. We show that fine-tuning with only a modest amount of structured data (64K sequences) can elicit these latent capabilities, transforming a standard video backbone into a pure visual reasoning engine. This reveals a valuable insight: complex generation can be achieved by leveraging the dynamic world priors inherent in video models, providing a practical alternative to current T2I paradigms.
> > >
> > > 2. **A New Dimension of T2I Test-Time Compute.** Approaches to inference-time compute in T2I generation typically rely on either external verifiers for quality assessment, or unified MLLMs to guide generation via textual CoT (shown in ***Figure 1 of the paper***). Our work explores a new, highly promising scaling dimension: investing inference compute into ***discrete cross-frame spatiotemporal evolution***. By allowing the model to explicitly refine visual content frame-by-frame, we achieve clear compositional improvements: simply allocating compute to a compact 3-frame reasoning trajectory consistently improves the base model's performance from 0.55 to 0.86 on GenEval and from 5.939 to 7.468 on Imagine-Bench. Importantly, CoF-T2I achieves these notable results entirely through ***intrinsic visual reasoning***, independent of existing paradigms. This demonstrates that such sequential visual reasoning provides a practical and competitive strategy for inference-time optimization.
> > >
> > > [1] Video models are zero-shot learners and reasoners. arXiv 2025.

---

### Official Review · Reviewer_34vG · 2026-03-09

**Soundness:** 3
**Presentation:** 4
**Significance:** 4
**Originality:** 3
**Overall Recommendation:** 5
**Confidence:** 5

**Summary:**

The paper introduces CoF-T2I, a T2I model that leverages a pre-trained video model as a visual reasoner, generating a 3-frame latent Chain-of-Frame trajectory that progressively refines from coarse semantics to high-fidelity aesthetics and decoding only the final frame as the image. To train this behavior, the authors built a 64K CoF-Evol-Instruct dataset of 3-step "defective draft -> refined draft -> high-quality" visual trajectories constructed via a quality-aware pipeline and a unified editing primitive. Using a frame-wise VAE encoding scheme and Rectified Flow training on these trajectories, CoF-T2I shows superior results over several models on GenEval and Imagine-Bench metrics, with ablations showing that explicit intermediate supervision and independent frame encoding are crucial to the gains.

**Compliance With Llm Reviewing Policy:**

Affirmed.

**Final Justification:**

rebuttal addressed main concerns and reinforced assessment

**Key Questions For Authors:**

1) Why explicitly use 3 as a number in Chain-of-Frame trajectory? Any justification? Is it possible to ablate the number?

2) It might make sense to compare with Video Models that generate images e.g. Cosmos, or T2V models and slicing a representative frame (first or last) as image output.

3) How is the inference time and memory compared to standard T2I models?

4) If possible, would like to see performance compared to close-source model (e.g. nano-banana) under a small subset to check how closer the gap we are getting.

**Limitations:**

no. Maybe its worth mentioning the memory and inference time overhead as limitations, also identify the failure cases.

**Strengths And Weaknesses:**

S1) First work to apply video CoF in T2I context. Creative solution on both dataset construction and training recipe.

S2) The paper is clear to read, figures are easy to understand for each key components.

W1) The pick of 3 as a number in Chain-of-Frame trajectory feels arbitrary without proper justification.

W2) Missing inference time and memory analysis. Using a large video backbone and predicting a 3-frame latent sequence almost certainly increases inference time and memory versus a comparable single-image model.

W3) Feels a bit unfair to mainly compare with other T2I models with image model backbone, as CoF-T2I uses a video model backbone. More video models need to be compared in T2I capabilities, or with some T2I models that distill from T2V models (e.g. Cosmos-Predict2-2B-Text2Image).

---

> ### Author Rebuttal · Authors · 2026-03-31
>
> We sincerely appreciate your valuable comments and address them below.
>
> ---
> > #### **Q1: Justification and Ablation of the 3-Frame Trajectory.**
>
> >
> We provide justifications and ablations below:
>
> 1. **Minimal Reasoning Decomposition:** A 3 frame sequence effectively decouples visual generation into ***discrete stages***: layout initialization, semantic correction, and aesthetic refinement. Fewer frames entangle semantic and aesthetic edits, while additional frames over-decompose defects, causing redundant computation without meaningful gains.
>
> 2. **Training Length Ablation:** We compare our default model against variants retrained on 1 frame (Target only SFT) and 2 frame (draft and final) sequences. As shown below, 3 frames yield the ***most significant boost*** in complex reasoning tasks like Position and Color Attribution.
>
>     |Frames|GenEval(Overall)|Position|Color Attr.|
>     |-|:---|:---|:---|
>     |1|0.81|0.73|0.59|
>     |2|0.83|0.78|0.63|
>     |3 (Ours)|0.86|0.83|0.71|
>
> 3. **Inference Extrapolation:** Retraining on longer trajectories is computationally infeasible during rebuttal. Instead, extending inference via the backbone's ***native temporal extrapolation*** shows performance saturating at 4 frames and degrading at 6 due to out-of-distribution semantic drift. While empirical, this suggests that the 3-frame setup effectively captures the necessary reasoning stages (draft, correct, polish) without incurring the instability and latency overhead of longer sequences.
>
>     |Inference Frames|GenEval|ImagineBench|Latency(s)|
>     |:---|:---|:---|:---|
>     |3 (Ours)|0.86|7.468|121.70|
>     |4|0.87|7.521|176.13|
>     |6|0.86|7.508|300.87|
>
> ---
> > #### **Q2: Missing inference time and memory analysis.**
>
> >
> We measured latency and peak VRAM on a single A100 GPU. CoF-T2I explicitly adopts the ***inference-time scaling paradigm***, deliberately trading time for complex compositional accuracy.
> 1. **Inference-Time Scaling:** Comparing CoF-T2I-14B against Target-only SFT-14B isolates the value of test-time compute. The added latency yields a ***substantial 0.713 gain*** on Imagine-Bench, suggesting that intermediate reasoning resolves complex semantics that single-step models miss.
>
>     |Method|Type|Latency(s)|VRAM(GB)|GenEval|Imagine-Bench|
>     |:---|:---|:---:|:---:|:---:|:---:|
>     |SD3-Medium|Standard Image|6.78|17.93|0.74|5.780|
>     |FLUX.1-dev|Standard Image|26.4|24.7|0.67|6.060|
>     |CoF-T2I-1.3B|Visual CoF|27.73|25.35|0.79|6.855|
>     |Target-only SFT-14B|Single Step|42.3|41.5|0.81|6.755|
>     |BAGEL-Think|Textual CoT|102.8|32.5|0.82|6.930|
>     |CoF-T2I-14B|Visual CoF|121.7|49.4|0.86|7.468|
>
> 2. **Cost-Performance at 14B:** CoF-T2I-14B substantially outperforms textual reasoning MLLMs like BAGEL-Think with only an ***18% latency overhead***. Operating purely visually potentially avoids cross-modal information loss, elevating output fidelity.
> 3. **Favorable Scaling at 1.3B:** For standard budgets, CoF-T2I-1.3B achieves higher performance than FLUX.1-dev with comparable latency and memory, validating the ***structural advantage*** of visual reasoning even at smaller scales.
>
> ---
> > #### **Q3: Comparison with Video Models and Video-to-Image Baselines.**
>
> >
> We expanded baselines to include video-native (Cosmos-1.0 Text2World [1]) and video-distilled (Cosmos-Predict2 T2I [2]) models. For fairness, video models are evaluated extracting their final frame under default settings:
>
> |Model|ModelType|GenEval|Imagine-Bench|
> |-|-|:-:|:-:|
> |Cosmos-1.0-Diffusion-7B-Text2World|Video (Last Frame Slice)|0.58|6.012|
> |Cosmos-1.0-Diffusion-14B-Text2World|Video (Last Frame Slice)|0.61|6.059|
> |Cosmos-Predict2-2B-Text2Image|Video-to-Image Distilled|0.83|6.250|
> |Cosmos-Predict2-14B-Text2Image|Video-to-Image Distilled|0.84|6.410|
> |CoF-T2I (Ours)|Video (CoF Fine-tuned)|0.86|7.468|
>
> Results show raw video backbones yield ***suboptimal T2I performance***. CoF-T2I outperforms even highly-optimized video-distilled baselines, particularly on complex compositional tasks (Imagine-Bench: 6.410→7.468).
>
> [1] Cosmos World Foundation Model Platform for Physical AI. arXiv 2025.
>
> [2] NVIDIA Cosmos Predict2 Model Reference (official documentation), 2025.
>
> ---
> > #### **Q4: Closed-source Comparison.**
>
> >
> We compared CoF-T2I and the proprietary Nano-Banana on TIIF-Bench [1] under the Long setting.
>
> |Model|Reas.|Attr.+Reas.|Style|Text|
> |-|-|-|-|-|
> |Nano-Banana|98.42|86.74|93.33|88.65|
> |CoF-T2I (Ours)|82.15|76.42|80.00|30.51|
>
> CoF-T2I achieves ***~85% of the commercial model's performance*** in core reasoning (Reas., Attr.+Reas.). The Text gap stems from the Wan2.1 backbone, which lacks typographic optimization.
>
> [1] TIIF-Bench: How Does Your T2I Model Follow Your Instructions? arXiv 2025.
>
> ---
> > #### **Q5: Failure Cases Analysis.**
>
> >
> CoF-T2I may exhibit the following limitations:
> * ***Complex spatial overlaps***: Local coherence occasionally overrides global layout.
> * ***Early detail convergence***: High-fidelity early drafts can resist semantic shifts.

---

> > ### Author Rebuttal · Reviewer_34vG · 2026-04-03
> >
> > Thanks for the detailed response. I will adjust my score.

---

> > > ### Author Response · Authors · 2026-04-03
> > >
> > > Thank you for your valuable reviews and recognition of our work!

---

### Official Review · Reviewer_sEDP · 2026-03-11

**Soundness:** 3
**Presentation:** 2
**Significance:** 3
**Originality:** 3
**Overall Recommendation:** 4
**Confidence:** 4

**Summary:**

The authors propose  CoF-T2I, which repurposes a pretrained video generation model for text-to-image generation via a 3-latent refinement process. The paper also contributed a 64K dataset of 3-frame trajectories with progressively improving image quality. The proposed approach is evaluated against the base video model and several unified MLLMs on two benchmarks, demonstrating improvements in semantic alignment and aesthetic quality.

**Compliance With Llm Reviewing Policy:**

Affirmed.

**Final Justification:**

The author provides many experiments to answer my question. I will keep my rating.

**Key Questions For Authors:**

See weakness

**Limitations:**

yes

**Strengths And Weaknesses:**

### Strengths
- Novel and intuitive formulation: Treating the temporal axis of a video model as an iterative refinement dimension is a compelling approach to enable inference‑time reasoning for t2i, where content is progressively refined frame by frame via temporal attention.
- Independent VAE is well‑motivated: It preserves the model’s multi‑frame reasoning while mitigating motion artifacts; it may also prevent errors from early latents from binding later refinements, which could strengthen the model’s ability to correct mistakes.
- Effective data construction pipeline: The planner–editor–verifier loop provides a principled mechanism to enforce prompt consistency and inter‑frame coherence.
- Reproducibility: Implementation details for independent VAE are provided in the appendix, which improves reproducibility.
- The method is compared with several strong unified MLLMs on GenEval and Imagine-Bench (Tab. 1).
- Clear ablation studies on intermediate supervision and independent encoding, together with trajectory-level visualizations of the three-frame latent modeling, provide convincing evidence supporting the proposed formulation.

### Weaknesses

- The paper states that the training data is constructed using Wan2.1, Qwen-Image and Nano-Banana for three tiers data (L-615). However, the main comparison only reports Wan2.1 (Tab.1).  A direct comparison with Qwen‑Image would more convincingly show the advantage of video‑based reasoning.
- The paper identifies two inference‑time reasoning paradigms: additional multimodal verifiers and unified MLLMs (L-40). However, the experiments only compare against the second category. It would be interesting to see comparisons with approaches using additional multimodal verifiers.
- The evaluation is concentrated on object‑centric cases. There is limited evidence on more challenging cases (e.g. text rendering, extreme counting, unusual compositions), which would additionally increase confidence.
- Clarification needed: The Target‑Only SFT (L-366) variant is described as training only on the final frame (F3) after removing intermediate frames. It is not clear whether this means a true single‑frame input, or whether F3 is duplicated to preserve the multi‑frame input format (e.g., 3‑frame or 5‑frame clips required by the VAE in Appendix C.2). This detail matters for a fair comparison with the multi‑frame CoF‑T2I setup.
- Latent-to-pixel decoding ambiguity:  The model only decodes the last latent frame (L-189). However, given the VAE’s temporal compression (1 latent → 4 frames), it is unclear whether all four frames are generated and which one is ultimately selected.
- Fig. 7(a) (upper‑left) shows a prompt–image mismatch.

---

> ### Author Rebuttal · Authors · 2026-03-31
>
> We sincerely appreciate your valuable comments. We found them extremely helpful in improving our manuscript. We address each comment in detail, one by one below:
>
> ---
> > #### **Q1: Comparison with Qwen-Image to Validate Video-Based Reasoning.**
>
> >
> Thanks for your advice! We provide a direct comparison to evaluate the advantages of the video reasoning approach.
>
> |Model|GenEval(Overall)|Imagine Bench(Overall)|Position(GenEval)|Multi Object(Imagine)|
> |-|:-:|:-:|:-:|:-:|
> |Qwen-Image|0.87|7.329|0.76|7.500|
> |CoF-T2I (Ours)|0.86|7.468|0.83|7.797|
>
> **Key Insights**
>
> 1. **Efficacy of Visual Reasoning:** CoF-T2I is initialized from Wan2.1, a significantly weaker base model with a GenEval score of 0.55. Reaching near parity with Qwen-Image on GenEval and surpassing it on Imagine Bench using a limited 64K dataset confirms the performance gains are fundamentally driven by the ***Chain of Frame reasoning architecture***, extending beyond simple data distillation.
> 2. **Compositional Superiority:** CoF-T2I demonstrates clear advantages in structural and relation sensitive tasks. It outperforms Qwen-Image in ***Position by 0.07*** and ***Multi Object by 0.297***, showing that iterative visual reasoning uniquely facilitates complex step-wise scene organization.
>
> ---
> > #### **Q2: Comparison with approaches using additional multimodal verifiers.**
>
> >
> Thank you for the insightful suggestion. To evaluate the additional multimodal verifiers paradigm, we implemented a Best-of-8 baseline using **Unified Reward** (UR)[1] across diverse backbones. Setting N=8 aligns with widely adopted sampling baselines to comprehensively probe the zero-shot generative ceiling.
>
> |Model / Paradigm|GenEval Overall|Imagine-Bench Overall|
> |-|:-:|:-:|
> |SD3-Medium|0.74|5.780|
> |SD3-Medium + UR (N=8)|0.78|6.051|
> |FLUX.1-dev|0.67|6.060|
> |FLUX.1-dev + UR (N=8)|0.72|6.420|
> |Wan2.1|0.55|5.939|
> |Wan2.1 + UR (N=8)|0.64|6.267|
> |**CoF-T2I (Ours)**|0.86|7.468|
>
> These results confirm that external verifiers are ***fundamentally bottlenecked*** by the original generation distribution. By leveraging continuous and progressive frame relationships, our model ***actively self-corrects*** semantic defects during the generation process, effectively surpassing the sampling-based distribution ceiling.
>
> Reference:
>
> [1] Unified Reward Model for Multimodal Understanding and Generation. arXiv 2025.
>
> ---
> > #### **Q3: Evaluation on additional challenging datasets.**
>
> >
> Thanks for your advice. We added new experiments on the highly challenging **TIIF-Bench** [1], focusing on its "Long" prompt setting to strictly test models under extreme complexity.
>
> As shown below, CoF-T2I achieves ***state-of-the-art performance*** in Reasoning (testing extreme counting) and Attr.+Reas. (testing unusual compositions), confirming the core advantage of multi-frame iterative refinement. While Style remains highly competitive, Text rendering is bounded by the ***inherent limitations*** of the Wan2.1 video backbone, which is not explicitly optimized for typographic precision.
>
> |Model|Reas.|Attr.+Reas.|Style|Text|
> |-|:-:|:-:|:-:|:-:|
> |SD3-Medium|74.07|70.34|76.67|20.83|
> |FLUX.1-dev|72.39|73.34|66.67|52.83|
> |BAGEL|76.77|70.07|83.33|33.94|
> |Janus-Pro-7B|79.07|70.84|70.00|33.83|
> |CoF-T2I (Ours)|82.15|76.42|80.00|30.51|
>
> Reference:
>
> [1] TIIF-Bench: How Does Your T2I Model Follow Your Instructions? arXiv 2025.
>
> ---
> > #### **Q4: Needed clarification on Target-only SFT variant.**
>
> >
> Thanks for pointing out! Target-only SFT uses a 1 frame input (F3). For fair comparison with our multi frame setup, we evaluate a duplicated variant where the final target frame is copied to form a 3 frame clip ***(F3, F3, F3)***.
>
> |Method|Input|GenEval|Imagine Bench|
> |-|-|-:|:-:|
> |Target-only SFT|(F3)|0.81|6.755|
> |Target SFT Duplicated|(F3, F3, F3)|0.80|6.738|
> |CoF T2I (Ours)|(F1, F2, F3)|0.86|7.468|
>
> Both baselines fall short of CoF T2I, confirming gains arise from the ***progressive reasoning trajectory*** rather than static final frame supervision.
>
> ---
> > #### **Q5: Regarding the latent to pixel decoding ambiguity and VAE temporal compression.**
>
> >
> Sorry for the confusion caused. We restrict the video VAE to the initial temporal window of the 1+4n scheme, enforcing a strict ***1 latent to 1 frame*** mapping. Thus, decoding the last latent yields ***exactly one deterministic image***.
>
> ---
> > #### **Q6: Prompt and image mismatch in Fig 7(a).**
>
> >
> Thanks for pointing out! This is an ***annotation typo***. The correct prompt is: 'A photo of an umbrella built from interlocking seashells.' We will correct this in the revision.

---

> > ### Author Rebuttal · Reviewer_sEDP · 2026-04-03
> >
> > The author provides many experiments to answer my question. I will keep my rating.

---

> > > ### Author Response · Authors · 2026-04-03
> > >
> > > Thank you for your careful and constructive reviews!

---

### Official Review · Reviewer_Vuw3 · 2026-03-12

**Soundness:** 2
**Presentation:** 3
**Significance:** 2
**Originality:** 3
**Overall Recommendation:** 4
**Confidence:** 4

**Summary:**

This paper proposes to utilize Chain-of-Frame reasoning to improve the text-to-image generation performance. To achieve this, they curate a CoF-Evol-Instruct dataset containing of CoF trajectories that model the generation process from semantics to aesthetics. Specifically, this process involves curating a diverse set of prompts, generating images with strong state-of-the-art T2I generators, measuring the quality of the generated images, and perform edits over the images to make either better or worse to form the reasoning chain. Then they fine-tune a video diffusion model for chain-of-frame reasoning, and to avoid interference between frames, they encode the reasoning trajectory with separate VAEs. Empirically, they found that the video generation model based reasoner achieves good performance on T2I tasks on GenEval and Imagine-Bench.

**Compliance With Llm Reviewing Policy:**

Affirmed.

**Final Justification:**

The rebuttal provides multiple experiments to address my main concerns (e.g., fair comparison with reasonable baselines, latency comparison, reasoning trajectory analysis)

**Key Questions For Authors:**

N/A

**Strengths And Weaknesses:**

Strengths:
1. This paper proposes a conceptually simple but impactful reframing: treating T2I as a short “video” refinement chain is an elegant way to induce intermediate, inspectable states. The intuition is interesting and could potentially provide interpretable intermediate states for better observing the models' performance/failure cases.
2. Ablations demonstrate the effectiveness of the proposed approach, where a simple target-only SFT doesn't work well, and the independent VAE encoding design is essential for the approach to work well.
3. The results demonstrating the performance improvement across frames are interesting, aligning well with the intuition.
4. This paper is well written and easy to follow.

Weakness:
1. Missing comparison with the data curation approach (i.e., generate the images with three SoTA models, verify which frame it belongs to, perform editing to enhance semantics/aesthetics to get the final frame). It's important to show that the fine-tuned model can achieve better performance than simply following the data curation approach to get the final frame.
2. The latency and cost of inference with a video diffusion model should be calculated and compared with other baselines on the benchmark.
3. Though claiming the intermediate frames can provide interpretability, no good analysis is included to support this claim (e.g., validate whether the intermediate frames correspond to consistent “subgoals” (layout → semantics → aesthetics).
4. The dataset is of limited size (i.e., 64K). The generalization to broader T2I scenarios is unexplored. Testing on harder and more diverse benchmarks will be helpful.

---

> ### Author Rebuttal · Authors · 2026-03-31
>
> We sincerely appreciate your valuable comments and address them one by one below.
>
> ---
> > #### **Q1: Missing comparison with the data curation approach.**
>
> >
> Thanks for your advice! To isolate the learned gains from the construction heuristics, we evaluate a ***Curation Baseline*** that applies the exact same data curation pipeline directly at inference time on the benchmarks.
>
> |Method|GenEval|||ImagineBench||
> |-|:---:|:---:|:---:|:---:|:---:|
> |Metric|Overall|Position|Color|Overall|Multi-Obj|
> |Curation Baseline|0.83|0.76|0.64|7.240|7.510|
> |CoF-T2I(Ours)|0.86|0.83|0.71|7.468|7.797|
>
> CoF-T2I consistently outperforms this baseline. The online curation pipeline relies on discrete VLM routing and sequential editing, making it highly susceptible to error cascading. By ***internalizing this progressive correction*** into a continuous generation process, CoF-T2I effectively circumvents the uncertainty and accumulated artifacts of repeated external module interventions.
>
> ---
> > #### **Q2: The latency and cost of inference with a video diffusion model.**
>
> >
> Thanks for pointing out! We measured latency and peak VRAM on a single A100 GPU. CoF-T2I explicitly adopts the ***inference-time scaling paradigm***, trading real time speed for complex compositional accuracy.
>
> |Method|Type|Latency(s)|VRAM(GB)|GenEval|Imagine-Bench|
> |:---|:---|:---:|:---:|:---:|:---:|
> |FLUX.1-dev|Standard Image|26.4|24.7|0.67|6.060|
> |CoF-T2I-1.3B|Visual CoF|27.73|25.35|0.79|6.855|
> |Target-only SFT-14B|Single Step|42.3|41.5|0.81|6.755|
> |BAGEL-Think|Textual CoT|102.8|32.5|0.82|6.930|
> |CoF-T2I-14B|Visual CoF|121.7|49.4|0.86|7.468|
>
> 1. **Inference Time Scaling:** Comparing CoF-T2I-14B against Target-only SFT-14B (evaluated in our paper, fine-tuned exclusively on final frames) isolates the value of test time compute. The added latency yields a ***0.713 absolute gain*** on Imagine-Bench, proving intermediate reasoning resolves complex semantics that single step models miss.
>
> 2. **Cost-Performance at 14B Scale:** CoF-T2I-14B substantially outperforms textual reasoning MLLMs like BAGEL-Think with only an 18% latency overhead. Operating purely visually potentially avoids cross modal information loss, elevating output fidelity.
>
> 3. **Architectural Efficiency at 1.3B Scale:** CoF-T2I-1.3B demonstrates ***competitive efficiency***. It outperforms the FLUX.1-dev baseline across benchmarks with comparable latency and VRAM, validating the structural advantage of visual reasoning even at smaller scales.
>
> ---
> > #### **Q3: Lack of analysis to support the interpretability of consistent "subgoals".**
>
> >
> We calculated the average scores of 1,000 held out reasoning trajectories using ***Unified Reward*** [1], a reward model that provides decoupled scoring for Semantics and Aesthetics.
>
> |Reasoning Step|Semantics|Aesthetics|
> |-|-|-|
> |F1 (Draft)|3.25|3.10|
> |F2 (Refine)|3.85(+18.5%)|3.28(+5.8%)|
> |F3 (Final)|3.92(+1.8%)|4.05(+23.5%)|
>
> *(Note: Scores are on a 1-5 scale. Percentages indicate relative improvements from the previous frame).*
>
> These results show a clear stage-wise pattern: from F1 to F2, ***semantics improves substantially*** while aesthetics changes little, indicating semantic correction and layout grounding; from F2 to F3, ***aesthetics improves markedly*** while semantics remains stable, indicating detail and visual refinement. This supports our claim that the intermediate frames correspond to interpretable and consistent subgoals.
>
> [1] Unified Reward Model for Multimodal Understanding and Generation. arXiv 2025.
>
> ---
> > #### **Q4: Concerns regarding the limited dataset size and generalization to broader T2I scenarios.**
>
> >
> We address both concerns below:
>
> The 64K size should be viewed as post-training supervision for reasoning induction, not as pretraining-scale concept learning, and is ***comparable to recent open T2I instruction-tuning datasets*** such as ShareGPT-4o-Image [1] and BLIP3o-60k [2]. To examine generalization beyond the training distribution, we additionally evaluate on TIIF-Bench [3] under the long-prompt setting as below.
>
> |Model|Reas.|Attr.+Reas.|Style|Text|
> |-|:-:|:-:|:-:|:-:|
> |SD3-Medium|74.07|70.34|76.67|20.83|
> |Flux.1-dev|72.39|73.34|66.67|52.83|
> |BAGEL|76.77|70.07|83.33|33.94|
> |Janus-Pro-7B|79.07|70.84|70.00|33.83|
> |CoF-T2I (Ours)|82.15|76.42|80.00|30.51|
>
> CoF-T2I achieves ***state-of-the-art performance*** in Reas. (extreme counting/logic) and Attr.+Reas. (unusual compositions), confirming the core advantage of multi-frame iterative refinement. It also remains highly competitive in Style. Text rendering is lower, which is objectively bounded by the inherent limitations of the Wan2.1 video backbone rather than our reasoning framework.
>
> [1] ShareGPT-4o-Image: Aligning Multimodal Models with GPT-4o-Level Image Generation. arXiv 2025.
>
> [2] BLIP3-o: A Family of Fully Open Unified Multimodal Models—Architecture, Training and Dataset. arXiv 2025.
>
> [3] TIIF-Bench: How Does Your T2I Model Follow Your Instructions? arXiv 2025.

---

> > ### Author Rebuttal · Reviewer_Vuw3 · 2026-04-02
> >
> > The rebuttal provides multiple experiments to address my main concerns (e.g., fair comparison with reasonable baselines, latency comparison, reasoning trajectory analysis)

---

> > > ### Author Response · Authors · 2026-04-02
> > >
> > > Thank you for acknowledging our rebuttal and efforts!

---

### Decision · Program_Chairs · 2026-04-30

**Decision:**

Accept (regular)

**Comment:**

In this paper, a method called CoF-T2I is proposed, which introduces a visual reasoning method (chain-of-frame) that treat text-to-image generation as a 3 frame video generation (e.g., first frame with semantic errors, second frame with minor issues, and third frame with high quality). Its contributions include a 64K CoF-Evol-Instruct dataset containing of CoF trajectories with progressively improving image quality,  and the frame-wise  independent VAE encoding design for the video generation model. Moreover, experiments are conducted to its superiority, by comparing the proposed approach against the base video model and several unified MLLMs on two benchmarks, GenEval and Imagine-Bench.

In the original reviews, reviewers raised concerns about lack of some comparison experiments, insufficient justification of some claim (e.g, "the intermediate frames can provide interpretability"), ablation study of components of the proposed method, generalization to other T2V backbone, some technical design choices (e.g., why 3 frames?).

After rebuttal, most concerns have been addressed, and all other reviewers gave positive scores, while only reviewer 2zMK still have concern about theoretical and methodological contributions. However, even reviewer 2zMK agrees the paper is at a borderline level.

I feel the idea of chain-of-frame for T2I is novel to some extent, and the CoF-Evol-Instruct dataset contribution and the methodology contribution like independent VAE encoding are sufficient and the experiment results are promising. So I recommend "Accept".